EMBO
Molecular Medicine

# A high-affinity, bivalent PDZ domain inhibitor complexes PICK1 to alleviate neuropathic pain

Nikolaj R Christensen[1,2,†], Marta De Luca[1,†], Michael B Lever[1], Mette Richner[3], Astrid B Hansen[1], Gith Noes-Holt[1], Kathrine L Jensen[1], Mette Rathje[1], Dennis Bo Jensen[4], Simon Erlendsson[5], Christian RO Bartling[2], Ina Ammendrup-Johnsen[1], Sofie E Pedersen[1], Michèle Schönauer[2], Klaus B Nissen[2], Søren R Midtgaard[6] (iD), Kaare Teilum[5], Lise Arleth[6], Andreas T Sørensen[1], Anders Bach[2], Kristian Strømgaard[2], Claire F Meehan[4], Christian B Vægter[3], Ulrik Gether[1] & Kenneth L Madsen[1,*] (iD)

## Abstract

Maladaptive plasticity involving increased expression of AMPA-type glutamate receptors is involved in several pathologies, including neuropathic pain, but direct inhibition of AMPARs is associated with side effects. As an alternative, we developed a cell-permeable, high-affinity (~2 nM) peptide inhibitor, Tat-$P_4$-$(C5)_2$, of the PDZ domain protein PICK1 to interfere with increased AMPAR expression. The affinity is obtained partly from the Tat peptide and partly from the bivalency of the PDZ motif, engaging PDZ domains from two separate PICK1 dimers to form a tetrameric complex. Bivalent Tat-$P_4$-$(C5)_2$ disrupts PICK1 interaction with membrane proteins on supported cell membrane sheets and reduce the interaction of AMPARs with PICK1 and AMPA-receptor surface expression *in vivo*. Moreover, Tat-$P_4$-$(C5)_2$ administration reduces spinal cord transmission and alleviates mechanical hyperalgesia in the spared nerve injury model of neuropathic pain. Taken together, our data reveal Tat-$P_4$-$(C5)_2$ as a novel promising lead for neuropathic pain treatment and expand the therapeutic potential of bivalent inhibitors to non-tandem protein–protein interaction domains.

**Keywords** biopharmaceuticals; calcium permeable AMPARs; maladaptive plasticity; scaffold proteins; synaptic plasticity
**Subject Categories** Neuroscience; Pharmacology & Drug Discovery

## Introduction

Excitatory communication between neurons in the central nervous system relies almost exclusively on glutamatergic neurotransmission. Concordantly, most neurological and psychiatric diseases, including devastating conditions such as neuropathic pain, feature distinct glutamatergic components, yet very few drugs targeting glutamate neurotransmission have been approved for clinical applications. Indeed, major efforts have been directed toward developing compounds targeting the ionotropic NMDA (N-methyl-D-aspartate) and AMPA (α-amino-3-hydroxy-5-methyl-4-isoxazolepropionic acid)-type glutamate receptors, but most such compounds have failed during clinical development due to lack of efficacy or as a result of unacceptable side effects (Tymianski, 2014).

An attractive alternative approach to manipulate glutamatergic neurotransmission is to target the synaptic scaffold proteins that orchestrate synaptic signaling complexes and dynamically regulate the surface expression and ion conductance of the ionotropic glutamate receptors in the postsynaptic density. Several of these proteins contain PDZ (PSD-95/Discs-large/ZO-1 homology) domains that are characterized by an elongated binding crevice, which binds the extreme C-terminus of interaction partners, including in several cases the ionotropic glutamate receptors themselves (Khan & Lafon, 2014). Despite the well-defined binding crevice, it has proven challenging to develop sufficiently potent small-molecule inhibitors of PDZ domain-mediated interactions as for protein–protein interactions (PPIs) in general (Laraia *et al*, 2015). Also, given the relatively low micromolar affinity of the interacting C-terminal peptides, it has likewise been difficult to develop efficacious peptide-based ligands.

1 Molecular Neuropharmacology and Genetics Laboratory, Department of Neuroscience, Faculty of Health and Medical Sciences, University of Copenhagen, Copenhagen, Denmark
2 Center for Biopharmaceuticals, Department of Drug Design and Pharmacology, Faculty of Health and Medicine, University of Copenhagen, Copenhagen, Denmark
3 Danish Research Institute of Translational Neuroscience (DANDRITE), Nordic-EMBL Partnership for Molecular Medicine, Department of Biomedicine, Aarhus University, Aarhus C, Denmark
4 Department of Neuroscience, Faculty of Health and Medical Sciences, University of Copenhagen, Copenhagen, Denmark
5 Structural biology and NMR Laboratory, Department of Biology, University of Copenhagen, Copenhagen, Denmark
6 Structural Biophysics, Niels Bohr Institute, University of Copenhagen, Copenhagen, Denmark
*Corresponding author. Tel: +45 23649401; E-mail: kennethma@sund.ku.dk
†These authors contributed equally to this work

An exception is a membrane-permeable peptide ligand targeting the postsynaptic scaffold protein PSD95 (Aarts *et al*, 2002), which recently passed phase 3 clinical trial for the treatment of ischemia after stroke (Hill *et al*, 2020). By generating a bivalent peptide ligand that simultaneously target two adjacent, "tandem" PDZ domains in PSD-95, it has been possible to dramatically enhance efficacy affinity/avidity (low nanomolar) and plasma stability (hours) (Sainlos *et al*, 2011; Bach *et al*, 2012).

PICK1 (protein interacting with C kinase) is another PDZ domain containing scaffold protein found in glutamatergic neurons (Hanley, 2008). PICK1 plays a central role in synaptic plasticity (Volk *et al*, 2010) and is a functional dimer with two PDZ domains flanking the central membrane binding BAR (Bin/amphiphysin/Rvs) domain, which also mediates the dimerization (Karlsen *et al*, 2015). The PICK1 PDZ domain interacts directly with the C-terminus of a number of different membrane proteins and kinases (Staudinger *et al*, 1997), including the GluA2 subunit of the AMPA receptors (AMPARs) (Dev *et al*, 1999; Xia *et al*, 1999). Functionally, PICK1 has been shown to regulate protein kinase C (PKC)-dependent phosphorylation of serine 880 (S880) of the AMPAR GluA2 subunit (Lu & Ziff, 2005), important for AMPAR trafficking during synaptic plasticity (Kim *et al*, 2001; Chung *et al*, 2003; Seidenman *et al*, 2003; Steinberg *et al*, 2006). Recent studies have further suggested a more direct role of PICK1 in regulating AMPAR surface stabilization, internalization, and recycling (Gardner *et al*, 2005; Jin *et al*, 2006; Cao *et al*, 2007; Rocca *et al*, 2008; Citri *et al*, 2010; Fiuza *et al*, 2017). In this context, expression of calcium permeable (CP) AMPARs in the process of homeostatic scaling (Clem *et al*, 2010) as well as in maladaptive plasticity associated with cocaine addiction (Wolf & Ferrario, 2010; Luscher & Malenka, 2011) and oxygen–glucose depletion (Dixon *et al*, 2009) has been shown to involve PICK1.

Finally, based on studies in animal models using inhibitory peptides, siRNA, and knock-out mice, PICK1 has been shown to be implicated in central sensitization of neuropathic pain (Garry *et al*, 2003; Atianjoh *et al*, 2010; Wang *et al*, 2011, 2016). This suggests PICK1 as a potential novel target for pharmaceutical intervention in chronic pain conditions where blockade of its PDZ domain can restrain unwanted sensitization by impairing glutamatergic AMPAR signaling (Garry *et al*, 2003; Atianjoh *et al*, 2010; Wang *et al*, 2011, 2016). Nonetheless, development of an efficacious inhibitor of the PICK1 PDZ has so far failed with the best inhibitors having affinities in the sub-micromolar to micromolar range (Garry *et al*, 2003; Bach *et al*, 2010; Thorsen *et al*, 2010; Marcotte *et al*, 2018).

Here, we present the development of a highly potent (1.7 nM) and efficacious peptide inhibitor of PICK1 for putative neuropathic pain treatment. Although the PICK1 PDZ domains in the functional PICK1 dimer are presumably > 150 Å apart (Erlendsson *et al*, 2015; Karlsen *et al*, 2015), we demonstrate that a bivalent peptide ligand displays a striking ~1,000-fold increase in affinity over analogous monomeric peptides. Size-exclusion chromatography and small angle X-ray scattering (SAXS) demonstrate that the bivalent peptide obtains avidity in a highly unique manner from the assembly of a novel complex of PICK1 dimers-of-dimers. We further show that the Tat-conjugated bivalent peptide is membrane permeable, engages with the target protein, and interferes with the PICK1-dependent phosphorylation of the GluA2 subunit of the AMPARs. Furthermore, the bivalent, but not the corresponding monovalent, high-affinity peptide can actively disrupt already established PICK1-receptor

complexes on supported cell membrane sheets, as well as interfere with PICK1-GluA2 co-immunoprecipitation and facilitate constitutive internalization of GluA2. Finally, we show that administration of the bivalent high-affinity peptide reduces spinal cord transmission and alleviates mechanical allodynia for up to 4 h in both the acute and chronic phase of the mouse spared nerve injury (SNI) model of neuropathic pain. Taken together, our results demonstrate that bivalent ligands represent a strong means to achieve high, pharmacologically relevant affinity even when targeting protein–protein interaction domains that only are adjacent to one another in transient oligomeric configurations of the target protein.

# Results

## Design and development of a high-affinity bivalent inhibitor of PICK1

Starting from the best-known binder of the PICK1 PDZ domain, DAT C13 (corresponding to the C-terminal 13 residues of the dopamine transporter (DAT)) ($K_i = 1$ μM) (Madsen *et al*, 2005), we sought to identify the shortest peptide sequence with conserved affinity toward the PICK1 PDZ domain. To this end, we took advantage of a fluorescence polarization (FP) competition assay using a fluorescent tracer peptide corresponding to the C-terminal 11 residues of the DAT (OrG-C11) (Fig EV1A). The DAT C13 (C13) competitor peptide was successively truncated from the N-terminus and retained affinity, down to DAT-C5 (C5), while further truncation (C4 and C3) slightly reduced affinity (Figs 1A and B, and EV1B). We evaluated the specificity of C5 (Fig EV1C) for a broad selection of PDZ domains (both class I and II) that were previously purified and had known ligands (Stiffler *et al*, 2007). C5 (10 μM) (black column) competed for binding to PICK1 as seen by the reduction in normalized mP (milli-polarization) compared to no peptide (full line). The C5 peptide primarily bound to the PDZ domain of PICK1, however with notable exceptions for Scribble (Scrb1) PDZ 2/4, Na(+)/H(+) exchange regulatory cofactor NHE-RF1 (NHERF) PDZ 2/2, and E3 ubiquitin-protein ligase PDZRN3 (Semcap3) PDZ 1/2, which were inhibited to similar level as for PICK1 (indicated by dashed line) (Fig EV1C). C5, however, did not compete with binding of either domain in full binding curves, suggesting they were false positive (Appendix Fig S1).

N-terminal modification of the C3-C5 peptides, similar to Bach *et al* (2011), did not significantly affect affinity (Appendix Table S1). To render the peptide cell permeable, we combined C5 with the 11-amino-acid cell penetration peptide from human immunodeficiency virus 1 (HIV-1) and trans-activator of transcription protein (Tat) (Richard *et al*, 2003; Lee *et al*, 2014) (YGRKKRRQRRR-HWLKV, termed Tat-C5). To be able to do binding studies, we also produced a fluorescent variant (5FAM-Tat-C5, Fig EV1E). Surprisingly, the addition of the Tat peptide gave rise to a dramatic increase in affinity of Tat-C5 for PICK1 ($K_i = 18.3$ nM), as assessed in competition with 5FAM-Tat-C5 (20 nM) (Fig 1C).

To explore the structural basis for the increased affinity, we took advantage of the PICK1 PDZ-DAT C10 construct (Erlendsson *et al*, 2014). Upon displacement of the DAT C10 (triggered by protease cleavage of the linker connecting the PDZ domain to the DAT C10 sequence) with equimolar amounts of the Tat-C5 peptide, we found

that residues in the actual binding groove I33, I35, and I37 in βB and L83, A87, and I90 in αB were mostly unaffected and that canonical PDZ binding is therefore not compromised by conjugation of

the Tat sequence. Instead, we observed strong chemical shift changes for several residues (G40, C44, C46, L47, Y48, and I49) in the Cys-loop, connecting βB and βC, as well as, a negatively charged

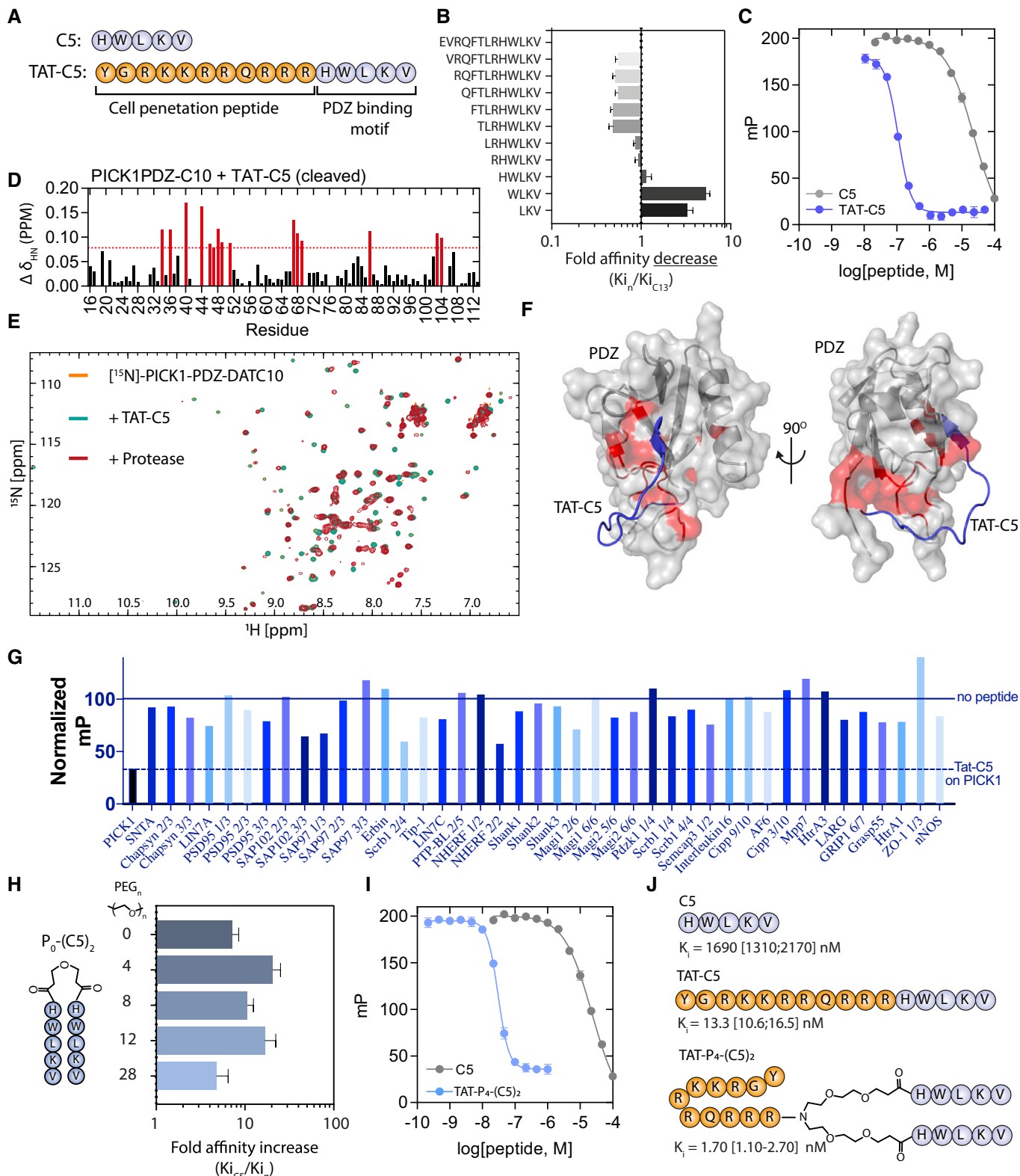

**Figure 1.**

◄

**Figure 1.  Development of a high-affinity bivalent inhibitor of PICK1s.**

A   Primary sequence of 5-mer peptide (C5) derived from the dopamine transporter C-terminus and the Tat-conjugated variant, Tat-C5.

B   Fluorescence polarization (FP) competition derived affinity assessment of consecutive truncations of the DAT C13 C-terminus as indicated by the sequences. Bars indicate the mean fold change relative to C13 ($K_i$ = 1.69 µM). Data are shown as mean with error bars as SEM with $n$ = 6. Representative binding curves are shown in Fig EV1A.

C   FP competition binding of C5 and Tat-C5 to PICK1 with 5FAM-Tat-C5 (20 nM) as tracer. Saturation binding curve is shown in Fig EV1B.

D   Backbone amide chemical shift changes ($\Delta\delta_{HN}$) of PICK1-PDZ-C10 after addition of Tat-C5 and addition of protease to cleave C10 from the PDZ domain and allow exchange. Numbers refer to residue number in PICK1 (UniProt: Q9EP80). Red bars indicate a chemical shift larger than mean+SD.

E   $^1$H-$^{15}$N-HSQC 2D spectra of PICK1-PDZ-C10 (orange) following addition of Tat-C5 (green) and subsequent addition of protease (red) to allow for exchange between C10 and Tat-C5.

F   Docking model, for visual purposes, of PICK1-PDZ (PDB: 2LUI) in complex with Tat-C5 (blue), with perturbed residues indicated from (D) in red.

G   Selectivity screen for Tat-C5 against a selection of 42 purified PDZ domains performed with a fixed concentration of Tat-C5 (10 µM) in competition with PDZ domains and their respective fluorescent ligands. Data are normalized to binding in absence of peptide (full line). Dashed line represents the level of competition obtained for PICK1. Screen for C5 can be seen in Fig EV1C.

H   Affinity gain of bivalent C5 peptides ($P_n$-(C5)$_2$) N-terminally conjugated with different length PEG linkers as indicated. Bars indicate the mean fold change relative to C5 ($K_i$ = 1.69 µM). Data are shown as mean with error bars as SEM with $n$ = 6. Representative binding curves are shown in Fig EV1C.

I    FP competition binding of C5 (also used in C) and Tat-P$_4$-(C5)$_2$ to PICK1 with 5FAM-Tat-C5 (20 nM) as tracer.

J   Structure and highest obtained affinity of C5, Tat11-C5, and Tat11-P$_4$-(C5)$_2$ toward PICK1.

Data information: Data points and bars are shown as mean with error bars as SEM of $n \geq 3$. Docking in (F) was done using HADDOCK (van Zundert et al, 2016), with residues indicated from (D) as essential residues for interaction, but no further restraints, therefore being only for visual purposes.

patch constituted by residues D68 and E69, opposite the canonical PDZ domain binding groove (Fig 1D and E). Docking the Tat-C5 (blue) using the observed chemical shifts as restraints (van Zundert et al, 2016) confirmed that canonical binding of the C5 residues allows for the electropositive Tat residues to wrap around the PDZ domain and simultaneously interact with D68 and E69 (Fig 1F). The specificity of Tat-C5 for the PICK1 PDZ domain increases when compared to the C5 peptide alone (Figs 1G and EV1C), which suggests that the exact spatial positioning of negatively charged residues and interaction with and/or flexibility of the Cys-loop are important for both affinity and specificity.

Based on our previous work (Bach et al, 2012), we next hypothesized that the native dimeric structure of PICK1 could be targeted using a bivalent ligand with two binding peptides connected via a polyethylene glycol (PEG) linker. We therefore designed a range of bivalent peptides of DATC5, which were fused at the N-terminus with PEG linkers of different lengths, PEG$_n$-(C5)$_2$ (Figs 1H and EV1D). All bivalent peptides showed > 10-fold increased affinity with an optimal linker length determined to be 4 PEG units, P$_4$-(C5)$_2$, spanning ~43 Å ($K_i$ = 98 nM) and giving rise to a 15-fold affinity increase compared to monovalent C5 ($K_i$ = 1.42 µM) (Figs 1G and EV1C). This was somewhat surprising given the average distance between the PDZ domains of the PICK1 dimer is estimated to be ~180 Å (Karlsen et al, 2015). The dimeric ligand with the longest linker (P$_{28}$-(C5)$_2$), which spans ~130 Å, on the other hand, showed the lowest affinity ($K_i$ = 593 nM), presumably due to increased entropic penalty (Figs 1H and EV1D). To render the bivalent peptide cell membrane permeable, we modified the PEG4 linker to enable conjugation to Tat (Bach et al, 2012), which again increased the affinity considerably (6.2 nM), as assessed in competition with 5FAM-Tat-C5 (20 nM) (Fig 1I). The resulting peptide Tat-PEG$_4$-(DATC5)$_2$, termed Tat-P$_4$-(C5)$_2$, was labeled with tetramethylrhodamine (TMR) (TMR-Tat-P$_4$-(C5)$_2$), and saturation binding curves also demonstrated low nanomolar affinity (Appendix Fig S2). Finally, no binding of Tat-P$_4$-(C5)$_2$ was observed to another bivalent target, PSD95, whereas an analogous dimeric inhibitor with a PSD95 specific sequence showed potent binding (Bach et al, 2009) (Fig EV1F).

Finally, using low concentration (2 nM) of 5FAM-Tat-C5 as optimal tracer (highest affinity) (Huang, 2003), we obtained final

affinities for Tat-C5 ($K_i$ = 13.3 nM/125-fold increase over C5) and Tat-P$_4$-(C5)$_2$ ($K_i$ = 1.7 nM/994-fold increase over C5) (Figs 1J and EV1G), respectively, ranking Tat-P$_4$-(C5)$_2$ among the most potent PDZ domain inhibitors. This affinity gain was ~15-fold from the bivalency and additional ~50-fold from the Tat sequence. Importantly, the peptides also potently competed with DAT and GluA2 tracers although this was challenging to assess due to inhibitor depletion in the assay (Appendix Fig S3).

Previous studies on the analogous PSD95 peptides have demonstrated that Tat conjugation as well as the PEG linker does not merely increase binding strength but also increases plasma stability, which is crucial for in vivo administration. Similarly, we observed that whereas the C5 peptide was completely degraded, ~5% of Tat-C5 and more than 50% of Tat-P$_4$-(C5)$_2$ endured incubation with human plasma for 24 h (Fig EV1H), making these peptides relevant for in vivo application.

## Tat-P$_4$-(C5)$_2$ dissociates PICK1 from membrane-embedded receptors

PICK1 serves its functional role as a scaffold protein interacting via its PDZ domain with receptors, transporters, and ion channels embedded in the cell membrane. To determine the efficacy of Tat-P$_4$-(C5)$_2$ and Tat-C5 to interfere with PICK1 binding to membrane-embedded proteins, we took advantage of the supported cell membrane sheet (SCMS) approach (Erlendsson et al, 2019) (Fig 2A). Using this approach, we recently demonstrated that PICK1 interacts with membrane-embedded Tac-YFP-DAT C24 (the single transmembrane IL2 subunit fused to YFP and the 24 C-terminal residues of DAT) with a binding strength of ~50 nM (Erlendsson et al, 2019). First, we pre-incubated fluorescently labeled recombinant rat PICK1 (300 nM SNAP543-PICK1) with increasing concentrations of C5, Tat-C5, or Tat-P$_4$-(C5)$_2$ to compete for binding to SCMSs expressing Tac-YFP-DAT C24 (Fig 2B–D and Appendix Fig S4). The IC$_{50}$ values (C5 = 3.80 µM; Tat-C5 = 101 nM; Tat-P$_4$-(C5)$_2$ = 7.0 nM) were slightly weaker than the affinities obtained by FP competition binding, but confirmed the dimeric ligand as a superior inhibitor of PICK1. Next, we determined the ability of the peptides to dissociate pre-bound PICK1 from the SCMS. Following pre-incubation of

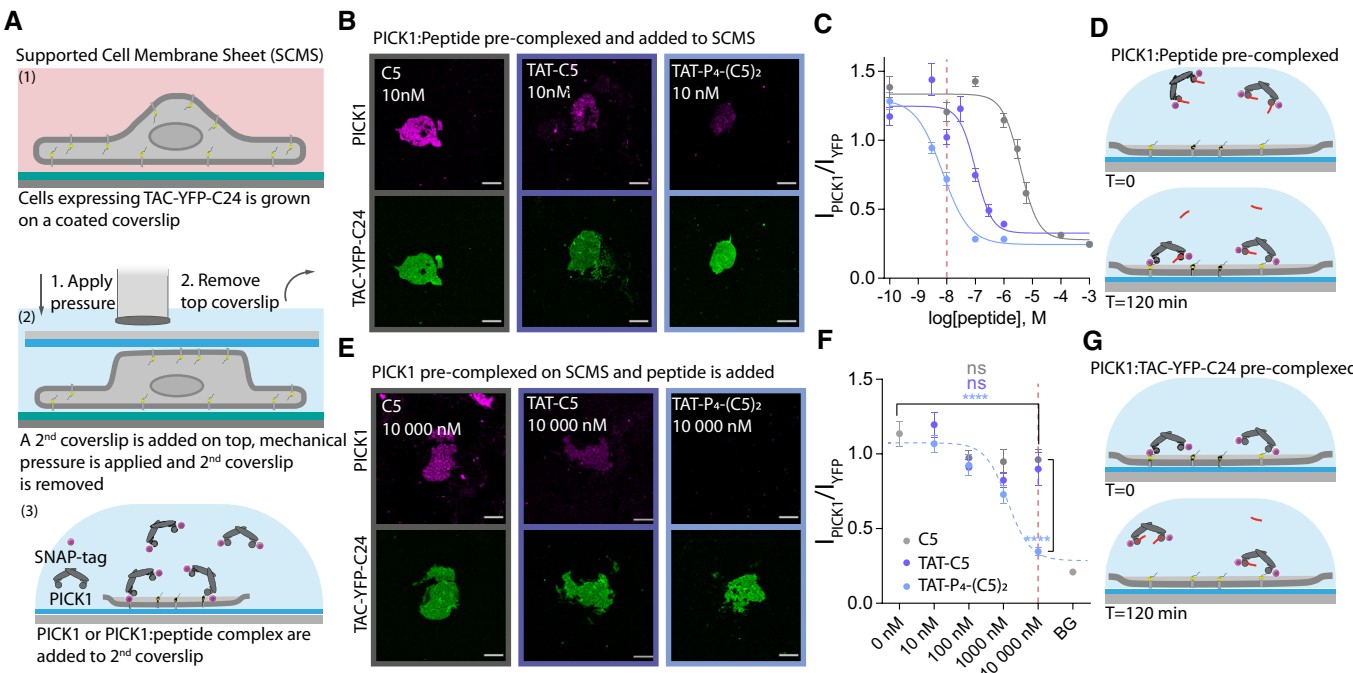

**Figure 2. High-affinity PDZ inhibitors can dissociate PICK1 from transmembrane interaction partner.**

A Illustration of supported cell membrane sheet (SCMS) assay.

B Representative confocal images of SCMS expressing Tac-YFP-DATC24 incubated with pre-complexed fluorescently labeled PICK1 and unlabeled peptides as indicated (10 nM), scale bar 10 μm. Full concentration curves shown in Appendix Fig S4 and include the representative images shown here.

C SCMS derived competitive binding curves for peptides as indicated, red line indicates 10 nM as shown in representative images.

D Experimental outline of pre-incubation with peptide.

E Representative confocal images of SCMS expressing Tac-YFP-DATC24, incubated with fluorescently labeled PICK1 and subsequently unlabeled peptide as indicated (10 μM). Scale bar 10 μm. Full concentration curves shown in Fig EV2 and include the representative images shown here.

F Efficacy of the peptides as indicated to facilitate dissociation of PICK1 from SCMS, red line indicates 10 μM as shown in representative images.

G Experimental outline post-incubation with peptide.

Data information: Data points are shown as mean with error bars as SEM of $n \geq 14$. Curves were fitted to a log dose response (three parameters) extracting the $IC_{50}$ using GraphPad Prism 7.0. Data points in (F) were compared at different concentrations using two-way ANOVA followed by Tukey's multiple comparison test (****$P < 0.0001$).

---

fluorescently labeled PICK1 on SCMSs expressing Tac-YFP-DAT C24, unbound PICK1 was washed away before incubation with increasing concentrations of C5, Tat-C5, or Tat-P$_4$-(C5)$_2$ (Fig 2E–G and images supporting the full curves and method in Fig EV2A–E). Strikingly, Tat-P$_4$-(C5)$_2$ increased the macroscopic off-rate of bivalently bound PICK1 with an apparent $IC_{50} = 1.17$ μM, whereas neither C5 nor Tat-C5 significantly dissociated PICK1 from SCMSs. PICK1 pre-binding to the sheet is likely to give rise to depletion of the inhibitors, which means that we can probably not translate the absolute potency obtained from the experiment to a neuronal setting. The relative difference, however, is striking and suggests a unique advantage of the dimeric ligand in a therapeutic context where the ability to dissociate a preformed complex predictably would be highly advantageous.

## High affinity of Tat-P$_4$-(C5)$_2$ results from complex assembly with tetrameric PICK1

We recently demonstrated that PICK1 forms elongated oligomers in solution using SAXS. Interestingly, the PDZ domains between the individual dimers were predicted to be in much closer proximity than

the two PDZ domains within the dimers (Karlsen *et al*, 2015). We therefore wanted to assess whether the dimeric Tat-P$_4$-(C5)$_2$ peptide potentially could stabilize higher order PICK1 complexes. In analytical size-exclusion chromatography (SEC), purified PICK1 eluted with a main peak at ~12 ml (Fig 3A, pink), which, according to our previous studies, corresponds to dimeric PICK1 (Karlsen *et al*, 2015; Madasu *et al*, 2015). The SEC profile was unchanged by incubation with Tat-C5 (Fig 3A). Upon incubation with Tat-P$_4$-(C5)$_2$, however, the main elution peak shifted ~1 ml toward a larger hydrodynamic radius far above the hypothetical increase of 3 or 6 kDa resulting from binding of 1 or 2 bivalent peptides, respectively (Fig 3B, light blue). Incubation of PICK1 with the fluorescently labeled TMR-Tat-P$_4$-(C5)$_2$ peptide showed overlapping curves of the 280 nm absorbance and the 546 nm fluorescence, confirming that the left-shifted peak indeed contained both PICK1 and the peptide, whereas the void peak (8 ml) showed no peptide binding (Fig 3C). To address the stoichiometry of the complex, we incubated a fixed amount of PICK1 (40 μM) with increasing concentrations of Tat-P$_4$-(C5)$_2$ (Fig 3D and E) and found that, at a molar ratio of 1:4 (Tat-P$_4$-(C5)$_2$:PICK1), all dimeric PICK1 (blue line) was shifted into the complexed state (green line), indicating that complex formation involves 1 bivalent peptide

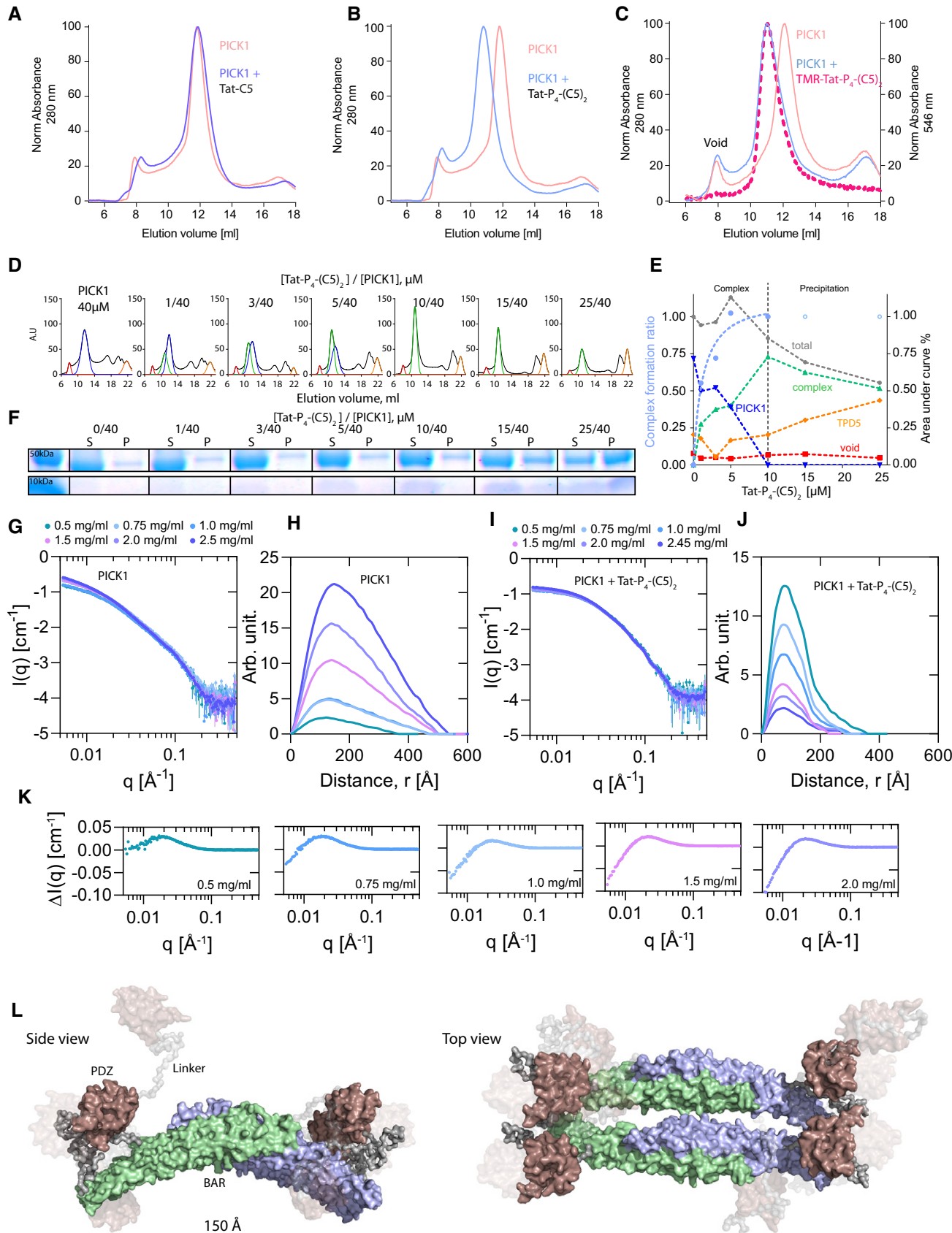

**Figure 3.**

**Figure 3. Tat-P$_4$(C5)$_2$, but not Tat-C5, reconfigures PICK1 into a novel tetrameric state.**

A  Size-exclusion chromatography (SEC) profiles of PICK1 in absence (pink, Abs$_{max}$ = 11.7 ml) and presence of Tat-C5 (purple, Abs$_{max}$ = 11.7 ml).

B  SEC profiles of PICK1 in absence (pink, Abs$_{max}$ = 11.7 ml) and presence of Tat-P$_4$-(C5)$_2$ (blue, Abs$_{max}$ = 10.7 ml).

C  SEC profiles of PICK1 in absence (pink, Abs$_{max}$ = 11.7 ml) and presence of TMR-Tat-P$_4$-(C5)$_2$ (blue (abs$_{280}$ nm), red (abs$_{544}$ nm), Abs$_{max}$ = 10.7 ml).

D  Representative SEC profiles of PICK1 (40 μM) incubated with increasing concentration of Tat-P$_4$-(C5)$_2$, colors indicate Gaussian fits of respective population, (red) aggregates/Void, (blue) dimeric PICK1, (green) oligomeric PICK1, (orange) unbound Tat-P$_4$-(C5)$_2$.

E  Quantification of abs$_{280}$ at E$_{vol}$ = 11.7 ml (dimer—blue) and E$_{vol}$ = 10.7 ml (tetramer—green) as a function of concentration (0-10 μM) of Tat-P$_4$-(C5)$_2$. Light blue line indicates tetramer fraction of total PICK1.

F  SDS–PAGE sedimentation assay showing increased amounts of PICK1 in the pellet (P) fraction with increasing concentrations of Tat-P$_4$-(C5)$_2$. PICK1~50 kDa, Tat-P$_4$-(C5)$_2$~10 kDa. S denotes supernatant fraction.

G  Small angle X-ray scattering (SAXS)-derived scattering curves of a concentration range of PICK1 on absolute scale show concentration-dependent aggregation.

H  Pair distance distribution function (pddf) of PICK1 in absence of Tat-P$_4$-(C5)$_2$, indicating very large protein complexes present in sample.

I  SAXS-derived scattering curves of a concentration range of PICK1 in complex with Tat-P$_4$-(C5)$_2$ on absolute scale show minor concentration-dependent aggregation.

J  Pair distance distribution function (pddf) of a concentration range of PICK1 in presence of Tat-P$_4$-(C5)$_2$, indicating a smaller complex than for PICK1 in absence of Tat-P$_4$-(C5)$_2$.

K  Scattering difference ($\Delta$I(q) = I(q)$_{PICK1+Tat-P4-(C5)2}$ − I(q)$_{PICK1}$) between PICK1 in presence and absence of Tat-P$_4$-(C5)$_2$ at indicated concentration shows a clear difference in scattering below 0.1 Å$^{-1}$.

L  Proposed EOM ensemble for PICK1, in complex with Tat-P$_4$-(C5)$_2$ (not visible in structure) with shading according to model ensemble percentage (10/20/70%). C-terminal unstructured regions are removed for clarity.

Data information: In (A), all elution profiles were normalized to the lowest and highest main elution peak. In (D), elution profiles were normalized to the lowest and highest points of PICK1 in absence of Tat-P$_4$-(C5)$_2$. In (G and H), the pddf was fitted using BayesApp. EOM fits were done using known structures of PICK1 PDZ domain (PDB: 2LUI) and the previously published model for the dimeric BAR domain, connected by full flexible linkers, N- and C-terminal. Model in (L) was prepared using PyMoL.

Source data are available online for this figure.

and two PICK1 dimers. At peptide to protein ratios above 1:4, the complex started to precipitate and could be pelleted by centrifugation (Fig 3F). To assess whether the ability of Tat-P$_4$-(C5)$_2$ to form higher order PICK1 complexes was dependent on the dimeric assembly of the BAR domain, PICK1 was dissociated into a monomeric state using detergent (0.1% Triton X-100) (Karlsen *et al*, 2015). Monomeric PICK1 eluted at 12.7 ml, and interestingly, a 1 ml shift was still observed upon incubation with Tat-P$_4$-(C5)$_2$. Moreover, Tat-P$_4$-(C5)$_2$ still maintained an affinity gain in FP binding of > 300-fold compared to C5, demonstrating that complex formation is independent of the dimeric BAR domain (Appendix Fig S5 and Appendix Table S2).

To elucidate the number of PICK1 subunits in the complex stabilized by Tat-P$_4$-(C5)$_2$ and to further investigate its structure, we used SAXS. Data were collected for a concentration range of 0.5–2.5 mg/ml for PICK1 both in absence and presence of Tat-P$_4$-(C5)$_2$ (Fig 3G–J and Appendix Table S3). The SAXS data and the corresponding pair distance distribution functions, *p(r)*, obtained by Bayesian Indirect Fourier transformations (www.bayesapp.org) for PICK1 alone (Fig 3H) changed considerably by incubation with Tat-P$_4$-(C5)$_2$ (Fig 3J), suggesting major conformational changes to the quaternary structure. The changes were most evident at low q/long distances (Fig 3K), indicating that the concentration-dependent large oligomer formation previously observed for PICK1 (Karlsen *et al*, 2015) was absent from the complex with PICK1:Tat-P$_4$-(C5)$_2$. Moreover, whereas the forward scattering, *I(0)*, of PICK1 without Tat-P$_4$-(C5)$_2$ showed a clear concentration dependence consistent with oligomerization as previously observed (Karlsen *et al*, 2015), *I(0)*, of the PICK1:Tat-P$_4$-(C5)$_2$ complex averaged ~4 PICK1 masses (229 kDa) across the PICK1 concentration range without any concentration dependence (Fig 3G–K, and Appendix Table S3). In conclusion, this demonstrates the formation of a stable, compact, tetrameric PICK1 complex by Tat-P$_4$-(C5)$_2$. Although structurally stabilized by Tat-P$_4$-(C5)$_2$, PICK1 was still flexible in solution (Appendix Fig

S6) and its shape cannot be represented by a single rigid structure (Karlsen *et al*, 2015). Consequently, we used ensemble optimization method (EOM) (Bernadó *et al*, 2007; Tria *et al*, 2015) on the 2 mg/ml dataset of PICK1 with Tat-P$_4$-(C5)$_2$ to investigate the structural organization of the complex. EOM sampling of multiple configurations of dimeric PICK1 fitted the data poorly (Fig EV3A–C), so we constructed tetrameric configurations of PICK1, including a circular (Fig EV3D–F), elongated (Fig EV3G–I), and a side-by-side configuration (Fig EV3J-L) and tested how they fitted the data. The most reliable fit of the SAXS data (simple model, low $\chi^2$) was obtained for a configuration of a compact tetrameric state (Fig EV3J–L), where the BAR domains align in a side-by-side configuration (Fig 3J) and the molecular model corresponding to this configuration is shown in Fig 3L. Taken together, the model independent SAXS analysis show that Tat-P$_4$-(C5)$_2$ induces a stable, compact, tetrameric state of PICK1 with the EOM analysis suggesting configurations with the PDZ domains from two individual PICK1 dimers within distances that can be bridged by the bivalent peptide.

## Tat-C5 and Tat-P$_4$-(C5)$_2$ are membrane permeable and interfere with hippocampal PICK1 function *ex vivo*

To confirm that the Tat sequence confers membrane permeability, dissociated hippocampal neurons were incubated with the TMR-labeled peptides TMR-Tat-P$_4$-(C5)$_2$, TMR-Tat-C5, or TMR-C5 together with the membrane dye DiO. Both of the Tat-fused peptides labeled neurons, whereas C5 alone did not (Fig 4A). Inspection of the 3D profile of the somatic region further revealed that the DiO staining, and thus membrane, surrounded a substantial fraction of the fluorescent signal from both TMR-Tat-P$_4$-(C5)$_2$ and TMR-Tat-C5, consistent with presence of the peptides in the cytosol (Fig 4A, boxed area highlighted in right panels). TMR-Tat-C5 in general, however, showed a more punctate distribution, whereas the TMR-Tat-P$_4$-(C5)$_2$ was mostly diffuse (Fig 4A).

To enable visualization of synaptic PICK1, we used a knockdown and replacement strategy (Citri *et al*, 2010) to exchange endogenous PICK1 with GFP-PICK1 in hippocampal neurons (Fig 4B). As previously shown, GFP-PICK1 colocalized with GluA2 containing AMPARs, both in the somatic region and in punctate

structures associated with the dendritic compartment (Fig 4B, zooms). Upon incubation (1 h) of the neurons with TMR-Tat-$P_4$-$(C5)_2$, we observed a punctate localization of the peptide signal that was associated with dendritic structures outside the somatic region and this signal showed partial overlap with the GFP-PICK1 and GluA2 signal

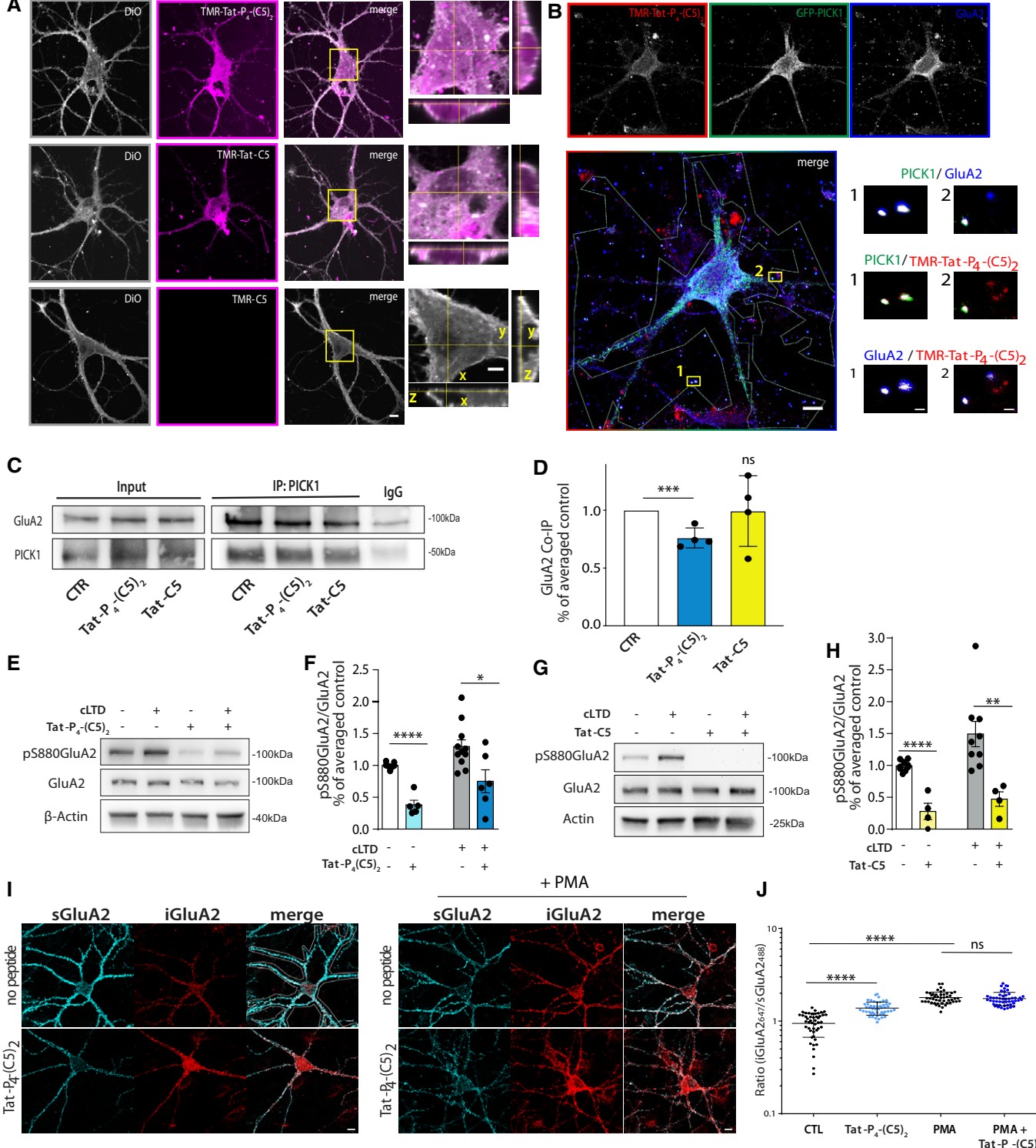

**Figure 4.**

**Figure 4. Tat-P$_4$-(C5)$_2$ is membrane permeable and compromise PICK1:GluA2 functional interaction in hippocampal neurons.**

A   Representative confocal images of hippocampal neurons showing membrane penetration of TMR-Tat-P$_4$-(C5)$_2$ and TMR-Tat-C5 (5 μM) (magenta), but not the control TMR-C5 (all 20 μM). The cell membrane was stained with DiO (gray). Scale bars: 10 μm and 5 μm on the zooms/orthogonal views.

B   Representative confocal images of hippocampal neurons transduced with the viral vector encoding GFP-PICK1 and incubated with 5 nM TMR-Tat-P$_4$-(C5)$_2$. Partial colocalization between TMR-Tat-P$_4$-(C5)$_2$ (red), GFP-PICK1 (green), and GluA2 (blue) is seen in insert zooms. Experiment was done three times. Scale bars: 10 μm and 1 μm on the zooms.

C   Representative immunoblots from hippocampal slices following treatment with Tat-P$_4$-(C5)$_2$ or Tat-C5 (bath application, 20 μM).

D   Densitometric analysis of immunoblots from (C) shows reduced PICK1:GluA2 Co-IP following incubation with Tat-P$_4$-(C5)$_2$ but not Tat-C5. Bar graphs show mean with error bars as SEM (one-way ANOVA followed by Dunnett's multiple comparison test, ***$P \leq 0.001$, $n$ = 4/group).

E   Representative immunoblots from hippocampal slices following Tat-P$_4$-(C5)$_2$ bath application (20 μM) show reduction in the phosphorylation level of S880 GluA2 both under basal condition and after chemical LTD (cLTD).

F   Bar graphs show densitometric analysis of immunoblots from (E).

G   Representative immunoblots from hippocampal slices following Tat-C5 bath application (20 μM) show reduction in the phosphorylation level of S880 GluA2 both under basal condition and after chemical LTD (cLTD).

H   Bar graphs (right panel) show densitometric analysis of immunoblots from (G).

I   Representative confocal images of hippocampal neurons stained for surface GluA2 (sGluA2, light blue) and internalized GluA2 (iGluA2, red). Treatment with Tat-P$_4$-(C5)$_2$ (20 μM) significantly increases the constitutive, but not PMA induced internalization. Scale bars 10 μm.

J   Quantification of iGluA2/sGluA2 within the region of interest for individual neuron (see I, top right corner). Tat-C5 did not increase GluA2 internalization (see Appendix Fig S9D–F).

Data information: Statistics in (F and H) was done by two-way ANOVA, followed by Sidak post-test, which revealed a significant effect of the peptide (F(1, 192) = 27,17), *$P \leq 0.05$, **$P \leq 0.01$, $n$ = 4–8 as indicated by individual values. Similar analysis in (J), PMA treatment (F(1, 192) = 217.6)) and a significant interaction factor (F(1, 192) = 40,39), ****$P \leq 0.0001$, $n$ = 46–50 neurons/group as indicated by individual values. All data expressed as mean with error bars as SEM.

(Fig 4B, zooms). These data demonstrate penetration of Tat-P$_4$-(C5)$_2$ into neurons and are consistent with subsequent *in vitro* target engagement of Tat-P$_4$-(C5)$_2$ with PICK1.

To further substantiate this conclusion, we knocked down PICK1 expression (without replacement) in the hippocampal neurons, which significantly reduced the amount of TMR-Tat-P$_4$-(C5)$_2$ signal in agreement with the reduced level of the target (Appendix Fig S7). We also observed clear colocalization of the TMR-Tat-P$_4$-(C5)$_2$ signal with GFP-PICK1 in HEK293 cells; however, this colocalization was neither seen for a PICK1 mutant with compromised PDZ binding (GFP-PICK1 A87L) nor with GFP alone (Appendix Fig S8A–I). The compromised binding of PICK1 A87L to TMR-Tat-P$_4$-(C5)$_2$ was confirmed by FP binding (Appendix Fig S8K). Finally, we were able to pull down GFP-PICK1, but not GFP-PICK1 A87L, from the transfected HEK293 cells using a biotinylated Tat-P$_4$-(C5)$_2$ (Appendix Fig S8J), directly supporting target engagement of TMR-Tat-P$_4$-(C5)$_2$ with PICK1.

To address if Tat-P$_4$-(C5)$_2$ or Tat-C5 could disrupt the interaction between PICK1 and AMPARs in neurons, we performed co-immunoprecipitation (co-IP) experiments. Incubation of hippocampal slices with Tat-P$_4$-(C5)$_2$, or Tat-C5, significantly reduced the co-IP of GluA2 by PICK1 (Fig 4C and D). Similarly, a reduction was observed in the reverse co-IP (Appendix Fig S9). Next, we addressed whether the peptides could interfere with the functional regulation of AMPARs by PICK1. Indeed, both peptides significantly reduced phosphorylation of Ser880 in GluA2 (Lu & Ziff, 2005), both under basal conditions and after a chemical long-term depression (LTD) protocol involving NMDAR-dependent kinase activation (Fig 4E–H). Finally, since PICK1 is believed to play a critical role in AMPAR trafficking, we addressed whether the peptides might affect GluA2 internalization in cultured hippocampal neurons. Somewhat surprisingly, Tat-P$_4$-(C5)$_2$ significantly increased constitutive internalization of surface labeled GluA2. Activation of protein kinase C (PKC) by PMA treatment further increased GluA2 internalization, which occluded the effect of Tat-P$_4$-(C5)$_2$ (Fig 4I and J). Importantly, we observed a similar effect on GluA2 trafficking upon shRNA-mediated knock-down of PICK1 (Appendix Fig S10A–C). In

contrary, Tat-C5 neither affected constitutive nor PMA induced GluA2 internalization (Appendix Fig S10D–F). In summary, these data support that both peptides are membrane permeable and reduce GluA2 S880 phosphorylation, but only Tat-P$_4$-(C5)$_2$ can disrupt the PICK1:GluA2 complex and increase constitutive internalization of GluA2.

## Tat-P$_4$-(C5)$_2$, but not Tat-C5, reduces functional interaction of PICK1 with AMPAR in the spinal cord

PICK1 has been proposed as a putative target for treatment of neuropathic pain (Garry *et al*, 2003; Wang *et al*, 2011, 2016). To address pharmacokinetic and pharmacodynamic properties *in vivo*, peptides were administered intrathecally (i.t.) in naïve mice. TMR-Tat-P$_4$-(C5)$_2$ and TMR-Tat-C5 (20 μM) were clearly visible in both the ventral and dorsal parts of the spinal cord 1 h after injection (Fig 5A). Moreover, TMR-Tat-C5 (Fig 5A, bottom) appeared to distribute more homogenously to the gray matter part of the spinal cord than TMR-Tat-P$_4$-(C5)$_2$ (Fig 5A, top). Administration of Tat-P$_4$-(C5)$_2$ also reduced co-IP of GluA2 by PICK1 from the spinal cord (Fig 5B), similar to our observation in hippocampal slices, whereas Tat-C5 did not significantly reduce GluA2 co-IP by PICK1 (Fig 5C), suggesting different pharmacodynamic/kinetic properties of the two peptides also in the spinal cord upon i.t. administration. Next, we tested the effect of both peptides on GluA2 Ser880 phosphorylation levels on spinal cord slices and, in contrast to hippocampal slices, Tat-P$_4$-(C5)$_2$, but not Tat-C5, significantly reduced GluA2 S880 phosphorylation (Fig 5D). To address putative effects on AMPAR surface levels, we performed surface biotinylation experiments on spinal cord slices. Neither peptides affected GluA1 surface levels, whereas both peptides gave rise to a small but non-significant reduction in surface GluA2 levels (Fig 5E). Lastly, the ability of Tat-P$_4$-(C5)$_2$ to engage the target protein *in vivo* following i.t. administration in naïve mice was confirmed by a pull-down experiment between PICK1 protein and an N-terminally biotinylated Tat-P$_4$-(C5)$_2$ peptide (Fig 5F).

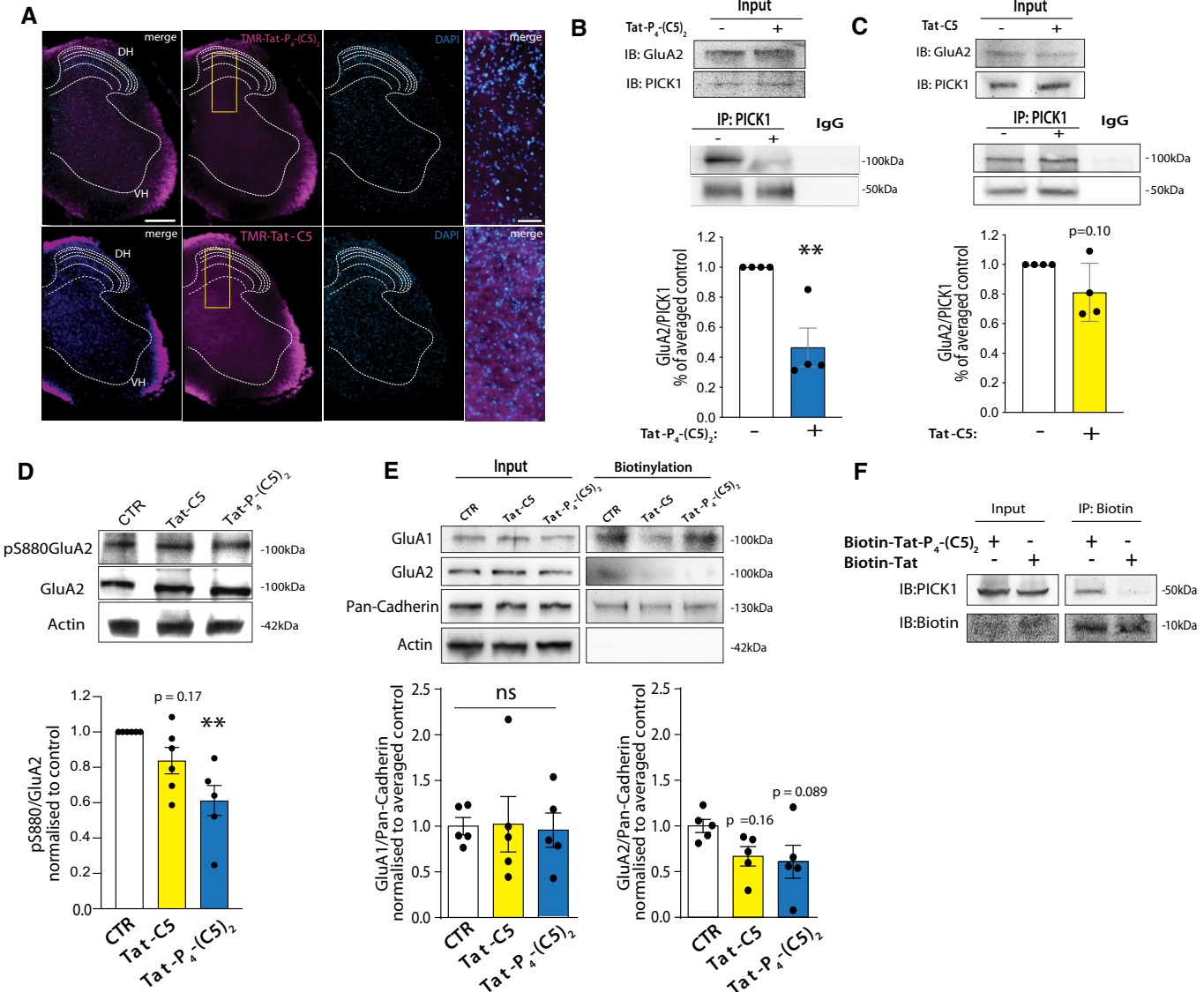

**Figure 5. Tat-P$_4$-(C5)$_2$ interferes with PICK1:GluA2 functional interaction in mouse spinal cord.**

A Representative coronal slices from lumbar spinal cord show the presence of both TMR-Tat-P$_4$-(C5)$_2$ and TMR-Tat-C5 peptides (*magenta*) 1 h after i.t. administration in naïve mice. Scale bar 200 µm. DH dorsal horn, VH ventral horn. Insert zooms of merged channels (right panels), scale bar 50 µm.

B Representative immunoblots of co-immunoprecipitated PICK1:GluA2 from spinal cord lumbar tract total lysates demonstrate partial disruption of this interaction 1 h after 20 µM i.t. injection of Tat-P$_4$-(C5)$_2$. Densitometric analysis of immunoblots indicates a significant effect following Tat-P$_4$-(C5)$_2$ treatment, and graph shows mean ± SEM and individual points, unpaired *t*-test, **$P \leq 0.01$, $n$ = 4/group.

C Representative immunoblots of co-immunoprecipitated PICK1:GluA2 from spinal cord lumbar tract total lysates demonstrates no disruption of this interaction 1 h after 20 µM i.t. injection Tat-C5. Densitometric analysis of immunoblots indicates no significant effect of Tat-C5, and graph shows mean ± SEM and individual points, unpaired *t*-test, $P$ = 0.10, $n$ = 4/group.

D Representative immunoblots of lumbar spinal cord total lysates exhibit decreased pS880-GluA2 total levels following 20 µM Tat-P$_4$-(C5)$_2$, and Tat-C5 i.t. injections compared to saline injected animals. Densitometric analysis of immunoblot indicates a significant effect only for Tat-P$_4$-(C5)$_2$ (graph show mean ± SEM and individual points, one-way ANOVA followed by Dunnett's multiple comparison test, **$P \leq 0.01$, $n$ = 5–6/group as indicated by individual values).

E Surface biotinylation of spinal cord slices under basal condition demonstrates unchanged GluA1 and GluA2 surface level upon Tat-P$_4$-(C5)$_2$, Tat-C5 peptides incubation compared to the untreated condition (CTR). Densitometric analysis of immunoblots shows that the reduction in mean surface for GluA2 in not significant (graph show mean ± SEM and individual points, one-way ANOVA followed by Dunnett's multiple comparison test, $n$ = 5/group as indicated by individual values).

F Immunoblot of PICK1 following pull-down from lumbar spinal cord with Biotin-Tat-P$_4$-(C5)$_2$ or Biotin following 20 µM i.t. administration demonstrating *in vivo* target engagement.

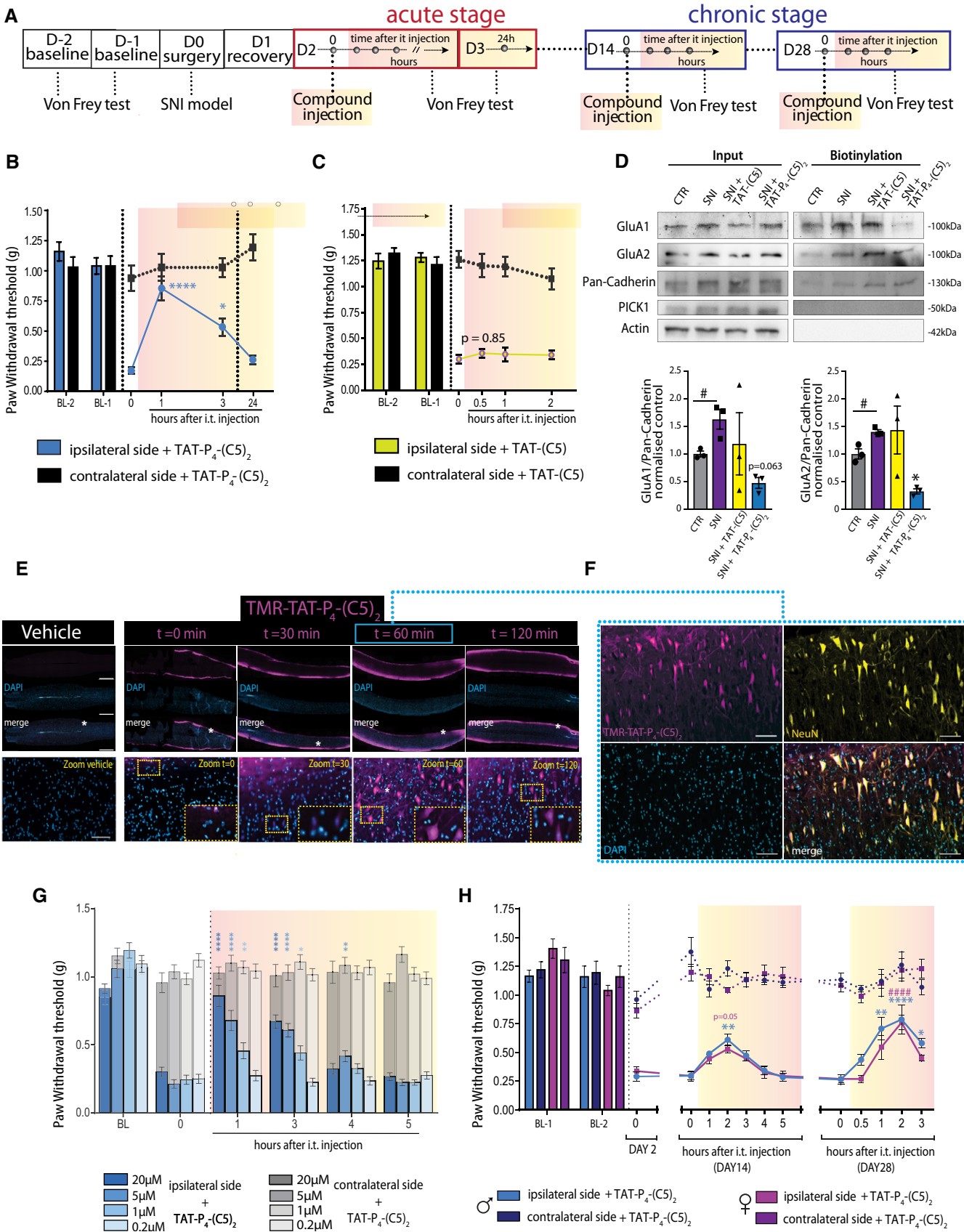

**Figure 6.**

◄

**Figure 6. Tat-P$_4$-(C5)$_2$, but not Tat-C5, reduce AMPAR surface expression and allodynia in SNI model of neuropathic pain.**

A  Diagram of *in vivo* timeline (D = day).

B  i.t. administration (20 μM) of Tat-P$_4$-(C5)$_2$ significantly reduced ipsilateral paw hypersensitivity of SNI mice in the initiation stage (2 days after injury) of the spared nerve injury model at time points 1 and 3 h (n = 7).

C  i.t. administration (20 μM) of Tat-C5 did not affect ipsilateral paw hypersensitivity of SNI mice in the initiation stage (2 days after injury) of the spared nerve injury model (n = 8).

D  Representative Western blots (top) for GluA1 and GluA2 following surface biotinylation of spinal cord slices. Densitometric analysis (bottom) of immunoblots. We observed upregulation of both GluA1 and GluA2 surface levels following SNI surgery compared to non-operated animal CTL (unpaired *t*-test GluA1: t(4) = 3.372, GluA2: t(4) = 3.991,#P ≤ 0.05, n = 3/group). Tat-P$_4$-(C5)$_2$ but not Tat-C5 significantly reduced the SNI-induced GluA2 surface expression and shows a strong tendency for GluA1 as well. Representative immunoblot shows unaltered PICK1 total level in the total lysate in the four conditions.

E  Images of lumbar spinal cord sagittal sections showing the time course of TMR-Tat-P$_4$-(C5)$_2$ (magenta) after i.t. administration (20 μM) in SNI mice showing maximal accumulation in cells after 60 min. Scale bar 1,000 μm and 100 μm in zooms.

F  Overlay with immunohistochemical staining for the neuronal marker NeuN (yellow) highlights strong neuronal tropism of TMR-Tat-P$_4$-(C5)$_2$ (magenta). Scale bar 100 μm.

G  Dose-dependent von Frey test in the acute phase (2 days after injury) of the spared nerve injury model following i.t. administration of Tat-P$_4$-(C5)$_2$ (n = 8/group) did not affect ipsilateral paw hypersensitivity of SNI mice and lasts up to 4 h post-injection.

H  i.t. administration of Tat-P$_4$-(C5)$_2$ (20 μM) at days 14 and 28 after injury in the maintenance phase of SNI model in mice shows significant increase in pain withdrawal threshold up to 3 h with equal efficacy in male and female mice (compared to time 0 pre-injection, D14 n = 8/group/gender, D28 n = 6 male, n = 7 female).

Data information: In (B and C), ipsi- and contralateral paw withdrawal thresholds at different time points were compared to time 0 using two-way repeated-measures ANOVA followed by Dunnett's multiple comparison test (*P < 0.05, ****P < 0.0001). In (D), GluA1/Pan-Cadherin levels were compared using one-way ANOVA followed by Dunnett's multiple comparison test, GluA2: F$_{(2,6)}$=6.256, *P ≤ 0.05; GluA1: F$_{(2,6)}$=2.843 P = 0.096, n = 3/group). In (G), ipsi- and contralateral paw withdrawal thresholds at different time points were compared to time 0 using two-way ANOVA followed by Dunnett's multiple comparison test (*P < 0.05, **P < 0.01, ****P < 0.0001). All data expressed as mean ± SEM. In (H), ipsi- and contralateral paw withdrawal thresholds at different time points were compared to time 0 using two-way repeated-measures ANOVA followed by Dunnett's multiple comparison test (*P < 0.05, **P < 0.01, ****/####P < 0.0001).

## Tat-P$_4$-(C5)$_2$, but not Tat-C5, reduces surface AMPAR levels to alleviate neuropathic pain

To test whether the PICK1 inhibitors can alleviate neuropathic pain symptoms that involve central sensitization, we employed the spared nerve injury model (SNI) on adult mice (Decosterd & Woolf, 2000; Fig 6A). On day two after the nerve injury, the model produced a robust hypersensitivity (reduced paw withdrawal threshold (PWT)) to mechanical stimuli applied to the ipsilateral hind paw without affecting the contralateral hind paw (Fig 6B). I.t. administration of Tat-P$_4$-(C5)$_2$ significantly alleviated SNI-induced hypersensitivity 1 h after administration (compared to the pre-injection time point, 0 h) reaching a level similar to both the contralateral paw and pre-injury level. The pain threshold was still significantly increased at 3 h (compared to the pre-injection time point, 0 h), while no effect of the peptide was seen after 24 h (Fig 6B). Intraplantar administration of the same dose did not affect PWT and even a 10-fold higher intraplantar dose did not significantly alleviate allodynia (Appendix Fig S11), suggesting little peripheral effect of Tat-P$_4$-(C5)$_2$.

In concordance with our previous results, i.t. administration of C5 did not affect the PWT, neither did Tat-C5 (Figs 6C and EV4A). Also, the small-molecule inhibitor of PICK1 FSC231 (Thorsen *et al*, 2010) as well as a myristoylated GluA2 inhibitory peptide (Garry *et al*, 2003) failed to elicit a significant effect on the PWT (Fig EV4B and C), whereas gabapentin, a first-line treatment for neuropathic pain, reversed PWT to baseline levels of the contralateral paw and prior to nerve injury of the ipsilateral paw (Fig EV4D). Surface biotinylation on spinal cord slices showed upregulation of both surface GluA1 and GluA2 following SNI, while treatment of SNI animals with Tat-P$_4$-(C5)$_2$, but not Tat-C5, reduced the surface level of both GluA1 and GluA2 below control level (Fig 6D).

Following i.t. administration in SNI animals, TMR-Tat-P$_4$-(C5)$_2$ initially (0 min and 30 min) distributed along the white matter of the spinal cord, and at later time points (60 min and 120 min), it was clearly visible in the gray matter as well (Fig 6E). We also noticed that the peptide signal in the gray matter was exclusively confined toward neurons (NeuN) (Fig 6F) and not on glial cells (GFAP) (Appendix Fig S12), suggesting neuronal tropism of TMR-Tat-P$_4$-(C5)$_2$.

Mice subjected to SNI showed similar alleviation of mechanical allodynia after the first (2 days after operation) and second (3 days after operation) administration of Tat-P$_4$-(C5)$_2$ (Appendix Fig S13) indicating no development of tolerance for Tat-P$_4$-(C5)$_2$ upon repeated administration. Using this setup, we could also demonstrate that the effect of Tat-P$_4$-(C5)$_2$ on PWT was dose-dependent (Fig 6G). Administration of Tat-P$_4$-(C5)$_2$ also significantly increased PWT of the animals at 14 days after injury and again at 28 days after injury (Fig 6H). Initial inflammation ceases 9–10 days after surgery, and therefore, these later time points are believed to better mimic chronic pain in humans (Percie du Sert & Rice, 2014). Injected gabapentin had comparable efficacy 14 days after nerve injury (Appendix Fig S14), suggesting that Tat-P$_4$-(C5)$_2$ may be an attractive lead for development of efficacious neuropathic pain therapy.

## Tat-P$_4$-(C5)$_2$ represses expression of CP-AMPARs and reduce neurotransmission in the dorsal horn of SNI mice

Both inflammatory and neuropathic pain models have been shown to cause an upregulation of CP-AMPARs (Vikman *et al*, 2008; Gangadharan *et al*, 2011; Chen *et al*, 2013). Since PICK1 has been shown to underlie maladaptive expression of CP-AMPARs in midbrain and hippocampus (Dixon *et al*, 2009; Wolf & Ferrario, 2010; Luscher & Malenka, 2011), we asked whether this might also be the case in the spinal cord. To assess functional expression of CP-AMPARs in neurons, we used kainate-induced cobalt uptake on spinal cord slices and as previously reported modest staining of cobalt was observed under basal conditions (Fig 7Ai and B) (Engelman *et al*, 1999). To mimic the effect of peripheral injury on spinal

cord slices, we applied TNFα which gave rise to robust induction of CP-AMPAR function (Fig 7Aii and B) (Xu *et al*, 2006; Wigerblad *et al*, 2017), and indeed, this induction was significantly decreased by co-incubation of TNFα with Tat-P$_4$-(C5)$_2$ (20 μM) (Fig 7Aiii and B), suggesting that Tat-P$_4$-(C5)$_2$ can interfere with functional expression of CP-AMPARs as one mechanism to reduce pain transmission.

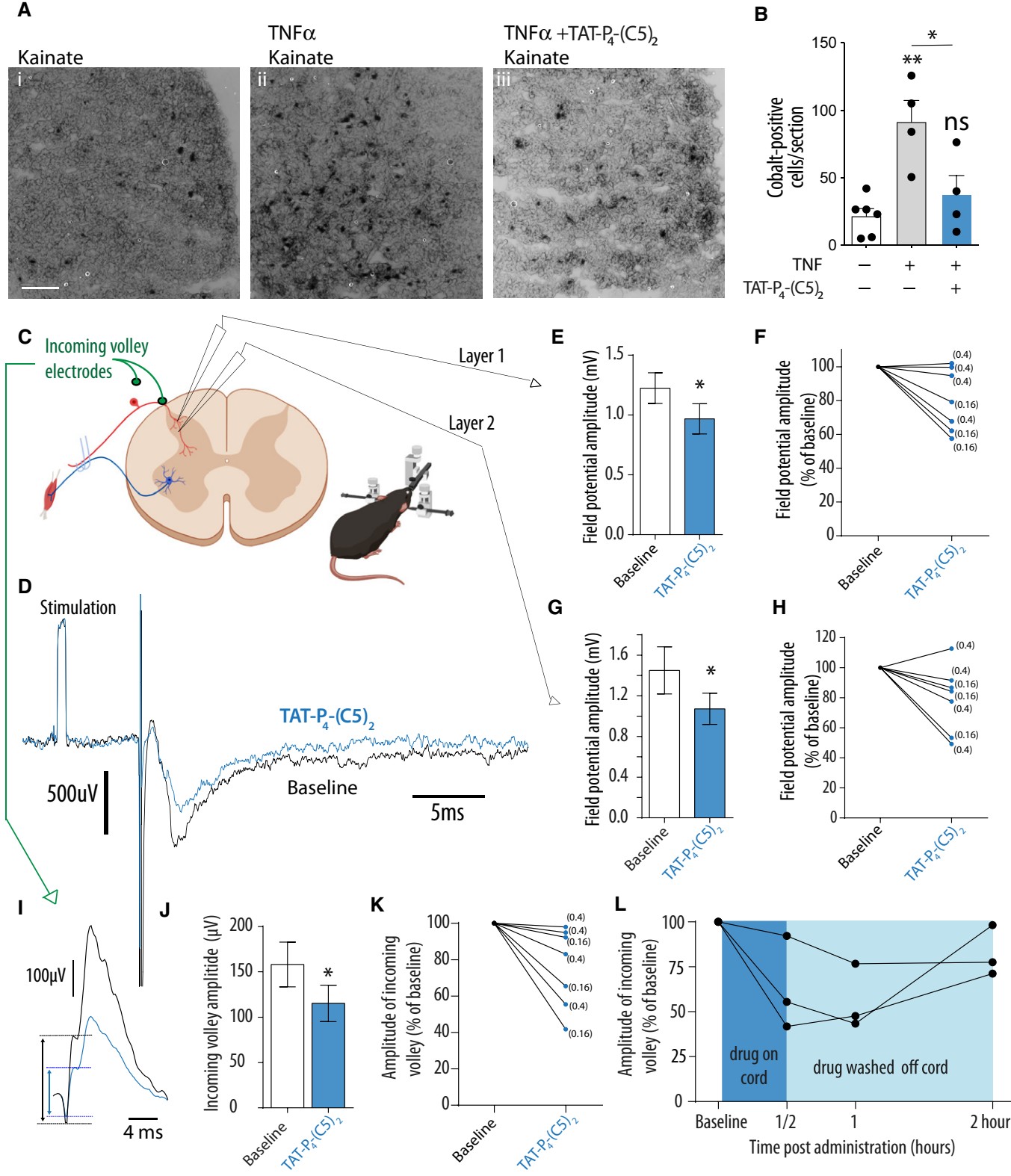

Figure 7.

**Figure 7. Tat-P$_4$-(C5)$_2$ reduces functional CP-AMPAR expression and electrophysiological transmission in the spinal cord.**

A   Representative images of kainate-induced cobalt uptake in spinal cord slices from p14 mouse pups treated with (i) saline, (ii) TNFα, and (iii) TNFα + Tat-P$_4$-(C5)$_2$. Scale bar: 100 μm.

B   Quantification of cobalt positive (black soma) cells. Each data point represents the average of 4-6 25-μm sections from a single 400 μm slice. Bars show means of 4–6 slices, and error bars represent SEM (one-way ANOVA followed by Bonferroni's multiple comparison test, *$P < 0.05$, **$P < 0.01$).

C   In vivo electrophysiological recordings for the dorsal spinal cord before and after peptide administration following stimulation of the peripheral sural nerve at intensities sufficient to activate C and Aδ fibers. Illustration of the experimental setup indicates where measurements were made (image made using Biorender.com).

D   Example of field potentials recorded in the same position before (black) and after (blue) peptide administration. The recording starts with a 1 mV calibration pulse followed by a stimulus artifact (S), which is truncated for the figure and finally the field potential at a depth corresponding to lamina 1. Each trace is an average of at least 10 successive trials.

E   Bar chart showing the mean amplitude of the field potential at lateral positions in lamina 1 recorded before and approximately 30 min after peptide administration, which was significantly reduced (Wilcoxon matched pairs, *$P = 0.0313$, $W = -26$, $n = 7$ mice, error bars: SEM).

F   Line chart to show data from (E), as separate data points for individual mice expressed as a percentage relative to baseline measurements. From this, it can be seen that the response was variable across mice with just over half showing a large reduction and the remaining mice only showing a relatively minor reduction (with no reduction in one mouse). The von Frey thresholds for the individual mice before the experiment are shown next to each line.

G   Bar chart showing the mean amplitude of the field potential at lateral positions in lamina 2 recorded before and 30 min after peptide administration, which was significantly reduced (Wilcoxon matched pairs, *$P = 0.0313$, $W = -26$, $n = 7$ mice, error bars: SEM).

H   Line chart to show data from (G), as separate data points for individual mice expressed as a percentage relative to baseline measurements. From this, it can be seen there was reduction in all but 1 mouse, although the magnitude of the reduction was again variable across mice. The von Frey thresholds for the individual mice before the experiment are shown next to each line.

I   Example of cord dorsum potentials recorded following stimulation of the peripheral sural nerve. The initial 3 inflections represent the incoming volley of action potentials along the afferent axons just before they enter the spinal cord. This was recorded before (black) and again 30 min post-peptide administration (blue). In this particular example, an extreme drop in the amplitude of the incoming volley is observed post-administration.

J   Bar chart showing that the mean amplitude of the incoming volley is reduced post-administration (Wilcoxon matched pairs, *$P = 0.0156$, $W = -28$, $n = 7$ mice, error bar: SEM).

K   Line chart to show data from (J), as separate data points for individual mice expressed as a percentage relative to baseline measurements. From this, it can be seen that the response was variable across mice with half showing a large reduction and the remaining mice only a minor reduction.

L   Amplitude of the incoming volley expressed as a percentage of the baseline recording prior to peptide administration. In these 3 mice, a reduction in the incoming volley at 30 min post-administration is seen. After this recording, the remaining peptide was removed by washing the spinal cord and two further recordings made at 1 and 2 h. Here, it can be seen that the incoming volley does not deteriorate further but stabilizes or starts to return toward baseline levels confirming that the initial reduction was not due to a deterioration in the health of the animal or damage to the roots.

Finally, to directly determine the effect of Tat-P$_4$-(C5)$_2$ on DRG excitability and transmission in the spinal cord, *in vivo* electrophysiological recordings were performed in adult decerebrate SNI mice to avoid possible interactions with anesthetics (Fig 7C and D). Hypersensitivity of the operated animals was confirmed by von Frey measurements prior to DRG excitability and transmission experiments (threshold for injured paw in individual mice is depicted in parenthesis Fig 7F and H). The sural nerve was stimulated and the incoming volley of action potentials along the DRG were recorded (Lloyd and McIntyre 1948), together with the resulting field potentials in the spinal cord at depths consistent with lamina 1 and 2 (Fig C, D).

Baseline recordings were made before Tat-P$_4$-(C5)$_2$ (20 μM) was applied to the exposed surface of the spinal cord and recordings performed again in the same location approximately 30 min after peptide application. Averaged field potentials in the spinal cord at depths corresponding to lamina 1 and lamina 2 were significantly reduced at 30 min following peptide administration relative to baseline measurements (Fig 7E and G) although there was significant variability in the magnitude of this effect between mice (Fig 7F and H). This confirms that the peptide reduced the net synaptic activity in the dorsal horn.

There was also a significant initial depression of the incoming volley observed at 30 min after administration (Fig 7I–K). In half of the mice, this was pronounced, as illustrated in Fig 7K. In the remaining mice, a smaller depression was observed. To confirm that this reduction was not due to damage to the dorsal roots or deterioration in the health of the mouse in three of the mice, the peptide was washed off the surface of the cord (Fig 7L). In all 3 mice, consistent with the time course in von Frey experiments, the incoming volley amplitude stabilized or started to return by 2 h, demonstrating that the effect was related to the action of Tat-P$_4$-(C5)$_2$, rather than a deterioration in the health of the animals of damage to the roots.

Taken together, these data demonstrate that Tat-P$_4$-(C5)$_2$ in addition to the modulation of central sensitization may also exert its action by suppression of excitability of the DRG axons at, or around, the dorsal root entry zone, which are likely significantly more accessible to the peptide than the secondary afferents in the spinal cord.

## Discussion

In this study, we develop two cell-permeable, high-affinity inhibitors, Tat-C5 and Tat-P4-(C5)$_2$, of the PICK1 PDZ domain with the purpose of disrupting the interaction of PICK1 with AMPARs and thereby reverse central sensitization and its related maladaptive plasticity believed to be associated with neuropathic pain. We demonstrate that the five C-terminal residues of DAT (called C5) are sufficient for maintaining affinity of the endogenous DAT ligand. Conjugation of C5 to the HIV-derived Tat sequence (Tat-C5), which rendered the peptide membrane permeably, serendipitously also increased affinity more than 100-fold—an effect that was specific for the PICK1 PDZ domain leading to increased selectivity. Our structural data suggest that this affinity gain results from interaction of Tat with residues extending from the binding groove and all the way to the opposite side of the PDZ domain.

In a parallel approach, we show that linking two C5 peptides together with a PEG linker result in a 15-fold affinity gain, while addition of the Tat sequence increased affinity further 50-fold

leading to more than 1,000-fold higher affinity for PICK1 than the C5 peptide alone. Indeed, bivalency is well known to endow biological molecules with high affinity/avidity and it is a well-known strategy in receptor pharmacology (Portoghese, 2001; Krishnamurthy *et al*, 2007) as well as in the case of the analogous bivalent inhibitor of PSD-95 (Sainlos *et al*, 2011; Bach *et al*, 2012). However, whereas these targets traditionally have been bivalent themselves, such as the tandem PDZ domain of PSD-95 or strict PDZ dimers (Paduch *et al*, 2007), we found that Tat-P$_4$-(C5)$_2$ works by bridging two PDZ domains from individual PICK1 dimers to produce a novel tetrameric complex. Importantly, although analogous to our previously published bivalent PSD-95 inhibitors, Tat-P$_4$-(C5)$_2$ does not bind PSD95, suggesting that its function in pain does not relate to the ability of PSD-95 to sustain hyperalgesia (D'Mello *et al*, 2011). Our finding implies that bivalent inhibitors may hold an unforeseen potential as high-affinity inhibitors—not only of tandem domains, but also of PDZ domain proteins with propensity to oligomerize, such as PDZ-GEF and LARG (Paduch *et al*, 2007).

Interestingly, the *in vivo* efficacy of Tat-P$_4$-(C5)$_2$ clearly exceeded the one of Tat-C5, as well as the efficacy of two other PICK1 inhibitors (myr-GluA2 and FSC231), with respect to relieving mechanical allodynia in the SNI model of neuropathic pain. Tat-P$_4$-(C5)$_2$ was likewise considerably better than Tat-C5 at; i) blocking the interaction between PICK1 and GluA2 according to co-immunoprecipitation analysis; ii) regulating GluA2 Ser880 phosphorylation, and iii) reducing AMPAR surface levels. It is possible that this in part may reflect better pharmacokinetic properties of Tat-P$_4$-(C5)$_2$ compared to Tat-C5. That is, although distribution and membrane permeability appeared to be better for Tat-C5 (Figs 4A and 5A), the *in vivo* stability of Tat-P$_4$-(C5)$_2$ is likely better than that of Tat-C5 as suggested by our plasma stability measurements (Fig EV1H). On the other hand, the differences observed in the surface biotinylation assay (Fig 6D), which was performed on spinal cord slices *ex vivo,* suggest that also the pharmacodynamic properties of the two compounds differ. This was specifically supported by the ability of Tat-P$_4$-(C5)$_2$, but not Tat-C5, to affect GluA2 internalization in dissociated hippocampal neurons.

From a mechanistic perspective, we propose that the differences in efficacy between the two peptides relate to the increased ability of Tat-P$_4$-(C5)$_2$ to dissociate a preformed cluster of PICK1 bound to membrane-embedded ligands as observed on SCMSs. For multivalent interactions, the macroscopic dissociation rate is composed several individual microscopic dissociation rates. In the first dissociation step, the dissociation rate is balanced by an association rate (rebinding) governed by the local concentration of the PDZ domain, rather than global concentration of PICK1, which give rise to avidity (Chi *et al*, 2010). Competitive inhibitors do not affect microscopic association or dissociation constants and, thus, cannot increase dissociation of monovalent interactions for which the macroscopic dissociation rate equals the microscopic one. For bivalent interactions, on the other hand, competitors may increase the macroscopic dissociation rate, given that it can effectively compete with association rate (rebinding) governed by the local concentration of the PDZ domain in this case. Clearly, Tat-P$_4$-(C5)$_2$ does this much more efficiently than both Tat-C5 (Figs 2E–G and EV2) despite only ~10-fold difference in in-solution binding strength. We propose that PICK1 may assemble in a tetrameric configuration, similar to our SAXS structure (Fig 3L), upon binding to membrane-embedded receptors.

In this scenario, peptide binding (monovalent as well as bivalent) will interfere with PDZ interaction with ligand in the membrane. We recently demonstrated, however, that PICK1 achieve most of the avidity, not only from multiple PDZ interaction at the membrane the avidity, but rather from interaction with the lipid membrane by the amphipathic helix lining the BAR domain (Herlo *et al*, 2018; Erlendsson *et al*, 2019) and likely also the CPC loop in the PDZ domains (Shi *et al*, 2010) (Fig EV5, black dashed circles). Interestingly, according to our SAXS structure (Fig 3L), binding of Tat-P$_4$-(C5)$_2$ restricts orientation of the two PDZ domains, which may sterically compromise these membrane-interacting motifs and thereby also the rebinding events they govern (Fig EV5, bottom). Since Tat-C5 will not restrict the PDZ orientation (Fig EV5, bottom), this could explain why only Tat-P$_4$-(C5)$_2$ facilitate the macroscopic off-rate.

Our trafficking experiments performed in cultured hippocampal neurons with Tat-P$_4$-(C5)$_2$ and shRNA suggest that PICK1 serves to stabilize GluA2 on the surface by saving it from constitutive internalization. This is in concurrence with previous findings in the PICK1 KO mice showing reduced extra-synaptic, surface GluA2 (Gardner *et al*, 2005); however, little work has been done on the role of PICK1 in constitutive internalization of AMPARs. Several studies, however, show that removal of or interference with PICK1 reduces activity-dependent internalization of GluA2 (Matsuda *et al*, 1999; Chung *et al*, 2000; Iwakura *et al*, 2001; Hanley *et al*, 2002; Seidenman *et al*, 2003; Bell *et al*, 2009). Although this might seem contradictory, we propose that the increased constitutive internalization upon interference with PICK1 might, at least in some cases, explain the reduced activity-dependent internalization, either due to fewer mobile surface receptors for activity-dependent endocytosis to act upon or because of normalization to different constitutive rates. Note that our data would indicate a compromised PMA-dependent internalization in the presence of Tat-P$_4$-(C5)$_2$ if normalized to the respective constitutive internalization with and without peptide (Fig 4H).

Interestingly, the surface biotinylation experiments suggest that while Tat-P$_4$-(C5)$_2$ only modestly decreases surface GluA2 in the spinal cord under basal conditions, it robustly reduces surface GluA1 and GluA2 after induction of neuropathic pain. Moreover, Tat-P$_4$-(C5)$_2$ reduced the TNF-induced increase in CP-AMPARs, supporting that Tat-P$_4$-(C5)$_2$ is capable of potently modulating the central sensitization often associated with neuropathic pain. Indeed, *in vivo* electrophysiology confirmed reduced transmission in layer 1 and layer 2 after stimulation of peripheral afferents; however, the measurements also revealed a robust decrease in the incoming presynaptic fiber volley in agreement with previous detection of PICK1 immunosignal in layers 1 and 2 of the spinal cord as well as in DRGs (Wang *et al*, 2011). Whether this reflects a direct effect on DRG excitability or indirect modulation by central neurons remains to be determined but regards such mechanistic speculations this finding is important when evaluating pharmacokinetic properties of putative pharmaceutical with PICK1 as target in pain.

The selective engagement of PICK1 in AMPAR surface expression in the spinal cord after SNI-induced plasticity makes PICK1 a very promising drug target. In analogy, PICK1 seems to serve a specific role in potentiating AMPAR function during acquisition and reacquisition of cocaine dependence (Bellone & Luscher, 2006; Famous *et al*, 2008; Schmidt *et al*, 2013; Jensen, 2017), suggesting that PICK1 is a promising target in addiction. Indeed, Tat-P$_4$-(C5)$_2$

strongly reduces cocaine reinstatement in a self-administering paradigm (Turner *et al*, 2020). Moreover, PICK1 has been evoked as a putative target in ischemia (Dixon *et al*, 2009), Alzheimer's disease (Alfonso *et al*, 2014), and Parkinson's disease (He *et al*, 2018).

In conclusion, we developed a high-affinity inhibitor toward the PICK1 PDZ domain, Tat-$P_4$-$(C5)_2$. Tat-$P_4$-$(C5)_2$ obtains its high affinity by cross-linking PDZ domains from individual PICK1 dimers and demonstrates favorable pharmacodynamic properties to potently alleviate neuropathic pain in both acute and chronic states, regardless of gender. We believe the peptide to be a promising lead compound and, with the proper formulation to enable appropriate administration, to be a strong candidate for extended pre-clinical studies leading to future clinical trials for neuropathic pain following peripheral nerve damage (Costigan *et al*, 2009).

# Materials and Methods

## Protein expression and purification

*Escherichia coli* cultures (BL21-DE3-pLysS) transformed with a PICK1 encoding plasmid (pET41) were inoculated in 50 ml LB with kanamycin overnight and transferred into 1 L LB medium with kanamycin and grown at 37°C until OD600 = 0.6. Protein expression was induced with 1 mM IPTG and grown overnight at 20°C. Bacteria were harvested and suspended in lysis buffer containing 50 mM Tris, 125 mM NaCl, 2 mM DTT (Sigma), 1% Triton X-100 (Sigma), 20 μg/ml DNase 1, and half a tablet of cOmplete protease inhibitor cocktail (Roche) pr. 1 l culture at pH 7.4. Suspended pellet was frozen at −80°C. The bacterial suspension was thawed and cleared by centrifugation (F20 rotor, 36,000 × *g* for 30 min at 4°C). The supernatant was collected and incubated with Glutathione-Sepharose 4B beads (GE Healthcare) for 2 h at 4°C under gentle rotation. The beads were pelleted at 3,500 × *g* for 5 min, and supernatant was removed and beads were washed 2 times in a wash buffer consisting of 50 mM Tris, 125 mM NaCl, 2 mM DTT, and 0.01% Triton X-100, pH 7.4. Washed beads were transferred to a PD10 gravity column and were washed additionally three times. Bead solution was incubated with thrombin overnight at 4°C under gentle rotation. Cleaved PICK1 was eluted on ice, and absorption at 280 nm was measured on a NanoDrop 2000, and protein concentration was calculated using Lambert–Beer's law ($A=\varepsilon cl$), $\varepsilon_{A280}$PICK1 = 32,320 $(cm*mol/L)^{-1}$. PDZ domains for selectivity test were purified as described previously (Stiffler *et al*, 2007).

## Peptide synthesis

Fluorescently labeled peptides were conjugated by either cysteine–maleimide in the case of Oregon Green peptides or N-terminal Ahx linkage in case of 5FAM labeling. All the FAM-conjugated and Oregon Green-conjugated peptides were purchased, respectively, from TAG-Copenhagen and Schafer-N. The peptide corresponding to the 5 most C-terminal amino acids of the dopamine transporter C5, HWLKV, as well as 5FAM-C5 5FAM-Ahx-HWLKV, Tat-C5 YGRKKRRQRRR-HWLKV, 5FAM-Tat-C5 5FAM-Ahx-YGRKKRRQRRR-HWLKV, and TMR-Tat-C5 TMR-Ahx-YGRKKRRQRRR-HWLKV, was purchased from TAG-Copenhagen.

Bivalent ligands $PEG_0$-$(HWLKV)_2$, $PEG_4$-$(HWLKV)_2$, $PEG_8$-$(HWLKV)_2$, $PEG_{12}$-$(HWLKV)_2$, $PEG_{28}$-$(HWLKV)_2$, and 5FAM-P4-$(C5)_2$ were synthesized in-house by solid-phase peptide synthesis as described in Bach *et al* (2009).

TMR-Tat-$P_4$-$(C5)_2$ (TMR-YGRKKRRQRRR-$N$PEG$_4$-$(HWLKV)_2$) and Tat-$P_4$-$(C5)_2$ (YGRKKRRQRRR-$N$PEG$_4$-$(HWLKV)_2$) were synthesized by Wuxi AppTec Co., Ltd. (China) accordingly. All peptides were purified or delivered to > 95% purity verified by UPLC-MS.

## Plasma stability

The peptides were dissolved to a final concentration of 50 μM in human plasma (3H biomedical, Uppsala, Sweden) containing 2% DMSO and incubated at 37°C. Aliquots (45 μl) were added directly to 26 mg guanidine hydrochloride (GnHCl), vortexed, and incubated for 2 min. 90 μl of 10% trichloroacetic acid (TCA) in acetone was then added, and the suspension was incubated at 5°C for 24 h.

The suspension was centrifuged at 18,000 *g* for 5 min. The supernatants were quantified by analytical C8 RP-HPLC and normalized to the compound concentration at T = 0. Ligand recovery was > 82%.

## Fluorescence polarization

Fluorescence Polarization (FP) saturation binding was carried out using an increasing amount of PICK1 incubated with a fixed concentration of fluorescently labeled peptides as indicated. Competition FP was done at a fixed concentration of PICK1 and indicated fluorescent tracer, against an increasing concentration of unlabeled peptide. In general, we sought to match tracer affinities to expected competitor affinity for optimal assessment of actual binding strength (Huang, 2003). After mixing, the 96-well plate (a black half-area Corning Black non-binding) was incubated for 20 min on ice, after which the FP was measured directly on an Omega POLARstar plate reader using excitation filter at 488 nm and long pass emission filter at 535 nm. The data were plotted using GraphPad Prism 8.3 and fitted to either a sigmoidal dose response for saturation experiments or One site competition for competition experiments. Kis were automatically calculated using the Cheng–Prusoff equation. All binding isotherms were repeated at least twice with different purifications of PICK1 and each time with 3 technical replicates. For selectivity screens, individual PDZ domains (42 domains + PICK1) with their respective fluorescent ligand were spotted in 96-well plates. FP was measured in parallel for plates incubated with buffer, 10 μM of C5 or Tat-C5.

## Nuclear magnetic resonance

Isotopically labeled PICK1 PDZ-DAT C10 was expressed and purified as described in Erlendsson *et al* (2014). The isolated unbound PICK1 PDZ domain is not stable in solution; however, extending the PDZ domain with a cleavable (PreScission C3 protease) linker followed by the 10 C-terminal residues of DAT yields a highly stable complex suitable for structural determination by NMR (Erlendsson *et al*, 2014). Cleaving the linker in absence of Tat-C5 produces only very minor structural perturbation. Chemical shift and/or intensity changes observed when cleaving the linker in presence of a saturating concentration Tat-C5 can be directly related to the displacement of C10 and subsequent binding of Tat-C5 peptide.

NMR experiments were recorded on a Bruker Avance III HD 600 MHz spectrometer with a QCI quadrupole resonance cryoprobe. All experiments were carried out at 25°C in 150 mM NaCl, 50 mM Tris, pH 7.4, 10% D2O, 0.02% NaN3 (Sigma-Aldrich), and 0.25 mM 4,4-dimethyl-4-silapentane-1-sulfonic acid (DSS) (Sigma-Aldrich). We used a PICK1 PDZ-DAT C10 concentration of 200 μM. For the competition binding experiments, we added Tat-C5 peptide to an equimolar concentration of 200 μM. For the PreScission protease (GE Healthcare)-induced competition binding experiments, we subsequently added 10 U (10 U/ml). In both cases, we monitored the change in amide chemical shifts by sequential acquisition of $^1$H-$^{15}$N HSQC spectra recorded using 8 scans and 128 increments.

The backbone chemical shifts for the construct were obtained by importing the peak list from the previous published NMR structure of the PICK1-PDZ-DAT C10 (Erlendsson *et al*, 2014) construct (BMRB entry: 18522, PDB ENTRY:2LUI). The imported peak list was adapted manually in CCPN analysis 2.4.1. The peak list were copied and adapted for each individual spectrum, and the chemical shift perturbation was quantified as the Euclidian distance ($\Delta\delta_{HN} = \sqrt{(0.5([\delta_H{}^2+(\alpha(\delta_N{}^2)])})$) between the center position of the original peak (before cleavage and addition of Tat-C5) and the position of the resulting peak after addition of Tat-C5 or Tat-C5 and C3 protease, respectively. Where $\Delta\delta_{HN}$ is the change in chemical shift in ppm, $\delta_H$ is the change in position of the peak in the 1H dimension, $\delta_N$ is the change in position of the peak in the $^{15}$N dimension, and $\alpha$ is a scaling factor between the two dimensions; in this case, 0.153 was used. The active peaks in the exchange between DATC10 and Tat-C5 were selected as being the residues with a chemical shift perturbation greater than two times the standard deviation of the mean chemical shift perturbation ($2\sigma = 0.077$).

The HADDOCK (High Ambiguity Driven protein–protein Docking) (van Zundert *et al*, 2016) algorithm was used to dock Tat-C5 into the PDZ domain of PICK1. Only, the chemical shift perturbation obtained was used as restraints in the docking. For the HADDOCK modeling, the following residues were chosen as active: I33,G34, I35, S36, I37, G40, C44, C46, L47, Y48, I49, Q51, G67, D68, E69, K83, V86, K103, L104, and the algorithm was provided with a the lowest energy structure of PICK1-PDZ-DAT C10 (PDB: 2LUI) where the DATC10 was removed prior to docking (final residues: 23-107). In Tat-C5 the residues, Y1, R3, R6, R7, Q8, H12, W13, L14, K15, and V16 were chosen as active residues. A total of 1,000 structures were initially docked, and the top 200 models were refined further using simulated annealing followed by water refinement. The best cluster of structures contained 42 structures, with a mean HADDOCK score of $-159.8 \pm 7.5$ arbitrary units with an overall RMSD of $0.8 \pm 0.4$ Å.

## HEK293-GT cultures and transfection

Human Embryo Kidney 293 GripTite cells (Thermo Fisher, catalog number: A14150) (HEK293-GT) for SCMS were grown in 75-cm$^2$ flasks until 70% confluent in Dulbecco's modified Eagle's medium 1965 with fetal calf serum and pen-strep antibiotics (DMEM 1965 + +). Cells were transfected with Tac-YFP-DAT C24 (pEYFP-C1 vector) (Erlendsson *et al*, 2019) using lipofectamine in opti-MEM® (Invitrogen) overnight. Cells were washed in PBS and detached using 0.5% Trypsin with EDTA (Sigma-Aldrich). Cells were counted in a Countess FL Automated Cell Counter (Thermo Fisher) using

Trypan blue (Sigma-Aldrich). Cells were seeded with a density of 200,000 cells pr. ml in a 6-well plate. Cells were then grown overnight at 37°C in a humidified 10% $CO_2$ atmosphere.

## Supported cell membrane sheets

The SCMS was prepared as described in Erlendsson *et al* (2019). In brief, round (Ø25 mm) coverslips (VWR 631-1346) were plasma cleaned (Harrick plasma cleaner) and coated with 0.3 mM poly-L-ornithine hydrobromide (Poly-ORN Sigma-Aldrich) for 30 min. Poly-ORN was washed out with 2 ml MilliQ water. The 6-well plates containing the cells expressing Tac-YFP-DAT C24 were washed twice in MilliQ for a total of 1 min, to allow cells to swell. The swelled cells were covered with a cover glass with the Poly-ORN-coated side facing down. Dynamic pressure was manually applied to the cover glass using the piston from a 12-ml plastic syringe for a total of 1 min. subsequent removal of the cover glass caused cell rapture leaving SCMS on the surface of the cover glass. The cover glass was covered in sheet buffer (10 mM HEPES, 120 mM KCl, 2 mM MgCl$_2$, 0.1 mM CaCl$_2$, and 30 mM glucose at pH 7.35) supplemented with 1 mg/ml BSA to block unbound Poly-ORN for 20 min on ice (in the dark). Coverslips were washed 3 times with sheet buffer. Protein solution (freshly prepared) was added and incubated according to the experiment. In the experiments using a premixed PICK1:peptide solution, 100 nM DY549-SNAP-PICK1 was incubated for at least 20 min with different concentrations of C5, Tat-C5, or Tat-P$_4$-(C5)$_2$ before addition to the cover glass. The protein:peptide solution incubated with the SCMS-cover glass for 2 h in the dark on ice. In the case of pre-binding of SCMS with PICK1, the SCMS was incubated with 400 nM of DY549-SNAP-PICK1 for 1 h on ice in the dark. Unbound protein was washed away with sheet buffer, and peptide solutions were added and incubated for 2 h in the dark on ice. The SCMS coverslips were then washed in sheet buffer and twice in PBS. The SCMS was fixed for 40 min in 4% PFA, washed 3× in PBS, and mounted onto objective glasses using Prolong Gold Antifade™ mounting medium (Life Science Technologies).

## SAXS

Purified recombinant PICK1 (40 μM) was incubated with Tat-P$_4$-(C5)$_2$ (20 μM). Precipitates were removed by centrifugation (5 min, at 16,100 *g* at 4°C). Sample composition was verified by size-exclusion chromatography (Superdex200, 10/300, 24 ml) to be homogeneous, and fractions were collected, pooled, and concentrated using a 10 kDa cutoff spin filter (Millipore) to a final concentration of 2.45 mg/ml. The PICK1 samples was prepared by isolating the dimer peak by collecting fractions from a HiLoad Superdex200 PG 16/600 (120 ml) and concentrating the dimer fractions to 3 mg/ml using a 10 kDa cutoff spin filter. Concentration series was prepared for both PICK1 and PICK1:Tat-P$_4$-(C5)$_2$, ranging from 0.5 mg/ml to 3 mg/ml and 0.5 mg/ml to 2.45 mg/ml, respectively. Samples were transported in an ice bath to the Deutsche Electron Synchrotron (DESY) facility, Hamburg, Germany. SAXS experiments (Appendix Table S3) were conducted at the DESY, Hamburg, Germany, using the P12 SAXS Beamline at the PetraIII storage ring (Blanchet *et al*, 2015). Preliminary data reduction includes radial averaging and conversion of the data into absolute scaled scattering intensity, *I(q),* as a function of the scattering vector *q*, where

$q = (4\pi \sin\theta)/\lambda$ (where $\theta$ is the half scattering angle) were done using the standard procedures at the beamline (Blanchet *et al*, 2015). As part of the process, the individual buffer and sample frames were looked through manually to check for radiation damage. The frames where indications of radiation damage were observed were removed, and the data averaging was done manually for the remaining files. For PICK1:Tat-P$_4$-(C5)$_2$, data merging was done by merging the low $q$-range data [0.01045–0.04036] Å$^{-1}$ from the 0.5 mg/ml sample together with the high q-range [0.03237–0.2694] Å$^{-1}$ data from the 2.45 mg/ml sample, with an overlap in the [0.03237–0.04036] Å$^{-1}$ $q$-range. The scattering data were merged, and buffer subtracted curves analyzed by Guinier analysis using the inbuilt analysis tools of the ATSAS program package (Petoukhov *et al*, 2012). The p(r) functions were plotted using http://bayesrelax.org/test/ with default settings. The shown data were binned using WillItRebin with default settings for clarity. Ensemble optimization method (EOM) (Bernadó *et al*, 2007; Tria *et al*, 2015) was used to fit the merged data using the all atom relaxed MD model of the dimeric PICK1 BAR domain from Karlsen *et al* (2015) together with the lowest energy structure of the PICK1 PDZ domain (PDB:2LUI) as fixed domains in the ensemble. The unstructured C-terminal, N-terminal, and PDZ:BAR linker were modeled using fully flexible dummy atoms and allowed for flexibility in the relative orientation of the domains. Tetrameric BAR domain models were created using PyMOL by duplicating, translating, and rotating the dimeric BAR domain into suggested conformations of the PICK1 tetramer (Karlsen *et al*, 2015). EOM (Ranch) (Bernadó *et al*, 2007; Tria *et al*, 2015) was used to generate 10,000 models for each BAR domain configuration with 15 harmonics. The models were fitted with either complete asymmetry or with 10% symmetric structures (in this case the 10% symmetric structures). EOM (Gajoe) (Bernadó *et al*, 2007; Tria *et al*, 2015) was used to fit the merged dataset to single pool (10,000 models) and multi-pool ensembles. Gajoe was run using 1,000 generations in the genetic algorithm with 100 ensembles, a non-fixed ensemble size, with a maximum of 20 curves pr. ensemble and 100 repetitions. Fits were evaluated on the basis of their $\chi^2$ value and their $D_{max}$ and $R_g$ distributions.

### Primary cultures of rat hippocampal neurons

Hippocampal neurons were prepared from prenatal E19 Wistar rat pups (mixed gender). Brains were isolated from 6 to 8 rat embryos, and brains were dissected in ice-cold dissection medium (HBSS (Gibco) supplemented with 30 mM glucose (Sigma-Aldrich), 10 mM HEPES (Gibco) pH 7.4, 1 mM sodium pyruvate (Gibco), 100 μg/ml penicillin, 100 μg/ml streptomycin (Sigma-Aldrich)). The cerebellum and meninges were removed, and hippocampi from both hemispheres were dissected out. The hippocampi were treated with papain (20 units/ml dissection medium, Worthington) for 20 min at 37°C, triturated (2 × 10 times) using differentially fire-polished Pasteur pipettes, and filtered through a 70-μM cell strainer. Dissociated neurons were seeded at a density of 50,000–100,000 cells/coverslip on poly-L-lysine (Sigma-Aldrich)-coated 25-mm glass coverslips emerged in Neurobasal medium (Gibco 21103-049) (supplemented with 2% (vol/vol) glutamax (Gibco 35050-038), 1% pen/strep, 2% (vol/vol) B27 (Gibco 7504-044), and 4% FBS (Gibco 7504-044)). Cell cultures were grown in a 37 °C, 5% CO$_2$, and 95% humidified atmosphere. After 24 h, the growth medium was substituted with serum-free medium, and cells were grown for 21 days, with addition of fresh growth medium every 3–4 days.

### Transduction of hippocampal neurons

At 14 days *in vitro* (DIV), lentiviral constructs packed with FUGWH1sh18eGFP (GFP-sh18) or FUGWH1sh18deleGFP (GFP) were added to each well. FUGWH1sh18eGFP expresses a short hairpin (sh18) that targets endogenous PICK1 and a shRNA-resistant eGFP-tagged PICK1, while the FUGWH1sh18deleGFP did not contain sh18 (Holst *et al*, 2013; Jensen, 2017). Lentiviruses were produced as described previously (Rasmussen *et al*, 2009).

### Immunocytochemistry on hippocampal neurons

21 DIV hippocampal neurons were incubated with 5 μM TMR-Tat-P$_4$-(C5)$_2$ or TMR-Tat-C5 or TMR-C5 for 1 h in conditioned media at 37°C, rinsed 3 times in PBS, and incubated with 5 μM of the membrane dye DiO (Thermo Fisher) for 10 min at room temperature. After an additional 3 washes in PBS, the hippocampal neurons were fixed in 4% PFA + 4% sucrose for 10 min at room temperature and subsequently washed three times in PBS and briefly in MilliQ. The coverslips were mounted by using Prolong Gold Antifade™ mounting medium (Life Science Technologies).

### *Permeability studies*

20-22 DIV hippocampal neurons were incubated with 5 nM of TMR-Tat-P$_4$-(C5)$_2$ for 1 h at 37°C, rinsed 3 times in PBS, and fixed in 4% PFA + 4% sucrose for 20 min (10 min on ice and 10 min at room temperature), rinsed again in PBS, permeabilized, and blocked in 0.05% Triton X-100 and 5% goat serum for 20 min at room temperature, labeled with primary GFP antibody for 1 h at room temperature, and followed by staining with goat anti-rabbit Alexa-488. After three final washes with PBS, the coverslips were mounted by using Prolong Gold Antifade™ mounting medium (Life Science Technologies). Quantification of the TMR-Tat-P$_4$-(C5)$_2$ peptide penetration was carried out by ImageJ software. Images (16 bit) were masked based on the 488-channel; the mask threshold was set at 1059 to 65535 and the background of the TMR-Tat-P$_4$-(C5)$_2$ at 726. The TMR-Tat-P$_4$-(C5)$_2$ peptide mean intensity within the mask was calculated for individual neurons, and each mean was divided by the corresponding masked area.

### Hippocampal neurons feeding assay

Live hippocampal neurons at 21 DIV were incubated with 20 μM Tat-P$_4$-(C5)$_2$ or Tat-C5 or vehicle for 1 h before internalization in conditioned medium with 2 μM TTX and 100 μg/ml leupeptin at 37°C. Surface AMPA receptors were labeled with the antibody anti-GluA2 (10 μg/ml) for the last 15 min. After 2 washes in PBS, neurons were treated with 1 μM PMA (Sigma-Aldrich) in conditioned media (with or without Tat-P$_4$-(C5)$_2$, Tat-(C5)) for 10 min. Neurons were fixed in 4% PFA + 4% sucrose for 5 min at room temperature, blocked with 10% goat serum for 15 min, and stained with anti-mouse Alexa-488 secondary antibody (5 μg/ml) in 3% goat serum for 45 min. Following fixation for 15 min, cells were permeabilized in 0.2% TX-100 for 5 min and stained with anti-

mouse Alexa-647 secondary antibody (1 µg/ml) in 3% goat serum for 45 min. The slides were washed three times in PBS and once in MilliQ and subsequently mounted to coverslips as described earlier. After outlining individual neurons on the basis of surface signal (488 nm), the mean intensity of the two channels was determined and the signal ratio (647 nm (internalized)/488 nm (surface)) was used as measure of internalization. A MatLab script was generated to automatically calculate overall intensity ratio for each neuron.

### Imaging

Imaging of SCMS, HEK293 cells, and hippocampal neurons was done on a Zeiss LSM 510 confocal laser-scanning microscope, equipped with an oil immersion objective, numerical aperture 1.4, 63× (Zeiss). For excitation of YFP, a 488 nm argon-krypton laser was used, with detection using a 505–550 nm bandpass emission filter. Red dyes were excited using a 543 nm helium-neon laser, and emission was detected using a 560–615 nm emission filter. Far red fluorophores were excited using a 633 nm laser light from a helium-neon (HeNe) laser source, and emitted fluorescent light was filtered using a long pass filter.

Images were acquired at $1,024 \times 1,024$ pixels, 8-bit pr. pixel, with 4-, 8-, or 16-line averaging scans. Images were treated using ImageJ or Zen (Zeiss software).

### Animals

SNI experiments were conducted on adult wild-type male mice (10–15 weeks old at initiation of experiment) from Taconic Biosciences, Denmark (C57BL/6JBomTac), unless otherwise stated. Mice were left to habituate for at least 2 weeks before starting the experiment.

Mice were housed with littermates in a temperature-controlled room maintained on a 12:12 light:dark cycle (tests performed during light phase) and allowed access to standard rodent chow Altromin1342 (Brogaarden, Denmark) and water *ad libitum*.

All experiments were approved by the Animal Experiments Inspectorate under the Danish Ministry of Food, Agriculture and Fisheries, and all procedures adhered to the European guidelines for the care and use of laboratory animals, EU directive 2010/63/EU. All efforts were made to minimize stress or discomfort as well as the number of animals used in each experiment.

### Co-immunoprecipitation

#### Acute hippocampal slices and lysates preparation

Acute hippocampal brain slices were prepared from adult male C57BL/6 mice (8–16 weeks old). The hippocampi were quickly dissected and sliced (400-µm transverse sections) into ice-cold aCSF buffer (124 mM NaCl, 3 mM KCl, 26 mM NaHCO$_3$, 1.25 mM NaH$_2$PO$_4$, 1 mM MgSO$_4$, 2 mM CaCl$_2$, and 10 mM D-glucose) and placed in carboxygenated aCSF for 1 h for recovery. Slices were then incubated with 20 µM Tat-P$_4$-(C5)$_2$ or Tat-C5 for 1 h followed by 20 µM NMDA for 3 min for chemical LTD induction, in aCSF buffer. Hippocampal slices were lysed in lysis buffer (50 mM Tris, 150 Mm NaCl, 0.1% SDS, 0.5% NaDeoxycholate, 1% Triton X-100, 5 mM NaF, and 1× Roche protease inhibitor cocktail at pH 7.4), and protein was incubated overnight at 4°C with 5 µg of antibody (either anti-PICK1 or anti-GluA2). Protein G agarose bead slurry was added

for 3 h, and the beads were then washed 3 times in lysis buffer. Proteins were eluted in 2× Laemmli sample buffer at 100°C for 6 min before Western blotting.

#### Spinal cord lumbar tract total lysates preparation

Spinal cord lysates were prepared from 10-week-old C57BL/6 mice. The animals were injected i.t. with 20 µM Tat-P$_4$-(C5)$_2$ or Tat-C5 and sacrificed 1 h post-injection. The spinal cords were dissected in ice-cold PBS1X by hydraulic extrusion according to the procedure described in Richner *et al* (2017). The lumbar tracts of the spinal cords were quickly harvested and lysed in lysis buffer. The same procedures described for hippocampal slices were used to produce spinal cord lysates.

#### Transfected HEK293 cells total lysates preparation

The same procedures described for hippocampal slices were used to produce HEK cell lysates.

### Surface biotinylation

Spinal cords were extruded from 8- to 12-week-old naïve and SNI mice and lumbar tract was dissected as in Richner *et al* (2011), and 400-µM transverse slices were generated using a McIlwain tissue chopper. Slices were then placed and separated in ice-cold aCSF (124 mM NaCl, 3 mM KCl, 26 mM NaHCO$_3$, 1.25 mM NaH$_2$PO$_4$, 1 mM MgSO$_4$, 2 mM CaCl$_2$, and 10 mM D-glucose) before recovery in carboxygenated aCSF for 1 h. Slices were incubated for 1 h with 20 µM peptide inhibitor and subsequently incubated in 1 mg/ml biotin (EZ-link sulfo-NHS-SS-Biotin: Thermo #21331) in ice-cold carboxygenated aCSF for 45 min, washed 3 times in 10 mM glycine, 1 time in Tris-buffer saline (TBS) (20 mM Tris, 150 mM NaCl, pH 7.6), homogenized in lysis buffer (25 mM Tris, 150 mM NaCl, 1% Triton X-100, 0.5% NaDeoxycholate, 0.1% SDS, 1 mM EDTA, 2 mM Naf, and 1 Roche protease inhibitor, pH 7.6) and centrifuged at 20,000 $g$ for 15 min. 500 µg of the supernatant was incubated with Streptavidin dynabeads (Life Technologies: 65601) for 2 h. Bead complexes were washed 3 times in lysis buffer and 1 time in TBS. Proteins were eluted in 2× Laemmli sample buffer at 100°C for 8 min before Western blotting.

### Spared nerve injury surgical procedure

Spared nerve injury (SNI) model was performed according to methods described in Richner *et al* (2011). Briefly, under isoflurane (2%) anesthesia, the skin on the lateral surface of the left thigh was incised and the biceps femoris muscle was divided lengthwise to expose the three branches of the sciatic nerve. The common peroneal and tibial branches were tightly ligated and axotomized distally to the ligation, while the sural branch was left intact. Animals were monitored daily for signs of stress or discomfort but in all cases recovered uneventfully.

### Drug preparation and administration

For intrathecal (i.t.) drug administration, 7 µl was injected into isoflurane anesthetized mice with a 10 µl Hamilton syringe (30 G) in the intervertebral space between L4/L5 or L5/L6 at an angle of 30–45° eliciting a Straub tail response as described previously (Hylden & Wilcox, 1980). For intraperitoneal (i.p.) injections, 10 µl/g per

mouse was administered with a 27 G needle. Drugs were diluted in 0.9% isotonic saline for both i.t. and i.p. injections.

### Behavioral experiments

Von Frey testing of mechanical allodynia was performed as previously described (Richner *et al*, 2011). In short, mice were placed in red plastic cylinders on a wire mesh (15-min habituation prior to testing). von Frey filaments (0.02–1.4 g) were applied in ascending order to the lateral part of the hind paws. Each von Frey filament was applied five times and a positive response in at least three out of five stimuli determined the threshold level. A positive response was defined as sudden paw withdrawal, flinching, and/or paw licking induced by the filament. Testing was performed blinded toward the nature of injected substance and by the same female researcher to avoid person-to-person variation and gender bias.

The paw withdrawal threshold (PWT) was estimated by using the following formula:

$$PWT = \frac{\text{Number of response failures}}{\text{Total number of trials}} \times ((\text{filament A} + 1gr)$$
$$- (\text{filament A}gr)) + \text{filament A}gr$$

### Fluorescence peptide administration

10-week-old SNI male mice were injected with 20 µM Tat-P$_4$-(C5)$_2$ i.t. and transcardially perfused with cold 0.1 M phosphate buffer (PBS) at time 0, 30, 60, and 120 min post-injection. *Naïve* animals were subjected to the same procedure after i.t. injection of 20 µM Tat-P$_4$-(C5)$_2$ or Tat-C5 and perfused 60 min post-injection. All spinal cords were isolated by hydraulic extrusion and post-fixed in PFA 4% for 8 h. Spinal cords were subsequently cryoprotected in 30% sucrose, embedded in Tissue-Tek® OCT, and sliced at 20 µm thickness by Cryostat Leica CM3050 S.

### Immunohistochemistry

Mounted slides were dried at room temperature, rinsed in PBS1X twice for 5 min each, and incubated in blocking buffer containing 5% goat serum, 1% BSA, 0.3% Triton X-100 in 1× PBS, for 90 min at room temperature. The slices were incubated in blocking buffer with primary antibody overnight at 4°C. On the second day, sections were rinsed 3 times in washing buffer (0.25% BSA, 0.1% Triton X-100 in PBS) 10 min each. Secondary antibodies Alexa-488 or Alexa-647 were diluted in washing buffer and the slides incubated 45 min at room temperature. After one wash in washing buffer and additional washing steps in 1× PBS, sections were air-dried and mounted with DAPI Fluoromount-G (Southern Biotech) mounting media.

Regarding the tissue distribution of Tat-P$_4$-(C5)$_2$ or Tat-C5, the slides were washed 5 min in PBS, air-dried, and mounted with DAPI Fluoromount-G mounting media.

### Epifluorescence microscopy

Epifluorescence microscopy was performed using a Zeiss axioscan.Z1, with a plan-apochromat 10×/0.45 objective (Carl Zeiss). LED light sources were used as follows, DAPI (353-nm excitation, 50-ms exposure, 460/70-nm bandpass filter), Alexa-488 (493-nm excitation, 50-ms exposure, 525/50-nm bandpass filter), TMR-568 (577-nm excitation, 100- to 300-ms exposure, 600/70-nm bandpass filter), and Alexa-647 (653-nm excitation, 200-ms exposure, 690/50-nm bandpass filter). All channels were imaged covering a total of around 6 µm of the 20-µm-thick spinal cord slices.

### Kainate-induced cobalt uptake

Postnatal day 14 (P14), C57BL/6 mouse pups of both genders were decapitated and their spinal cord was dissected out in ice-cold oxygenated stabilization buffer (139 mM sucrose, 32.5 mM NaCl, 2.5 mM KCl, 10 mM MgSO$_4$, 12 mM D-glucose, 24 mM NaHCO$_3$, 1 mM NaH$_2$PO$_4$, 0.5 mM CaCl$_2$). The lumbar part was embedded with cyanoacrylate in a 5% agar block, and 400-µm slices were cut on a vibrating microtome (Thermo Scientific Microm HM 650 V). The slices were divided into three groups and left to recover in oxygenated stabilization buffer at RT for minimum 60 min. Tissue from each animal was subjected to all three experimental groups. Slices were pre-incubated with 20 µM Tat-P$_4$-(C5)$_2$ (as indicated) in oxygenated stabilization buffer for the last 30 min of incubation in stabilization buffer. Slices were then transferred to oxygenated uptake buffer (139 mM sucrose, 57.5 mM NaCl, 5 mM KCl, 2 mM MgCl$_2$, 1 mM CaCl$_2$, 10 mM HEPES, 12 mM D-glucose) followed by addition of 20 µM Tat-P$_4$-(C5)$_2$ and 20 nM TNFα (as indicated) for 10 min. Co$^{2+}$ and kainate were added to a final concentration of 2 mM Co$^{2+}$ and 200 µM kainate for 20 min to all three experimental groups. Cobalt loading was terminated by 2 × 5 min wash in uptake buffer containing 3.5 mM EDTA followed by treatment with 0.12% (NH$_4$)$_2$S for 6 min to precipitate intracellular Co$^{2+}$. Slices were rinsed in PBS and fixed in 4% PFA for 3 h at 4°C followed by cryoprotection in 30% sucrose o/n at 4°C. Slices were rinsed in PBS, embedded in OCT, and cut to 25-µm slices on cryostat. Sections were processed to enhance the CoS reaction by silver intensification as described in Engelman *et al* (1999). Due to non-specific Co$^{2+}$ labeling in the outer parts of the 400-µm slices, these were excluded for analysis. Average cell count of 2–4 of the 25-µm slices from each 400-µm slice was used for quantification. Images were acquired using a Zeiss Widefield microscope using a 20× objective. For quantification, pictures were captured using identical exposure settings. Pictures were imported into ImageJ (version 2.0.0-rc-41/1.50b), and cells with black soma were counted manually. Analysis and counting were performed blinded by investigators.

### *In vivo* electrophysiology

Electrophysiological recordings were performed *in vivo* in adult decerebrate mice. The preparation has been previously described extensively (Meehan *et al*, 2012, 2017). Briefly, the initial surgery was performed under isoflurane anesthesia (1.5–2%), including the insertion of tracheal cannula for subsequent artificial ventilation, the dissection of the sural nerve (on the injured side), and a dorsal hemi-laminectomy between vertebrae levels T12-L1 (exposing spinal levels L3-L4). Mice were then placed in a modified Narishige frame and attached to a ventilator (CWE, 72 breaths/min). Craniotomy was performed, superficial blood vessels cauterized, and the entire brain rostral to the superior colliculus removed. The void was packed with surgical to prevent bleeding. After approximately

10 min, this was removed to confirm complete decerebration before being replaced followed by removal of isoflurane from the ventilation flow. Throughout the experiment, the temperature of the animals was kept stable at approximately 37°C using a heat lamp and heat blanket thermostatically controlled by a rectal thermometer. Expired $pCO_2$ was also stable throughout the experiments.

The dissected peripheral nerve was hooked on a custom-made bipolar stimulating electrode, and the spinal cord mechanically stabilized using vertebral clamps above and below the segments of interest. A silver ball electrode was placed on the lateral edge of the laminectomy, and a reference electrode placed down the side of the ribcage on the ipsilateral side. This recorded the cord dorsum potential that shows the incoming volley of sensory input in the dorsal roots.

The peripheral nerve was stimulated at thresholds to ensure maximal stimulation of the high-threshold Aδ and C fibers (Steffens *et al*, 2012). This electrode was manually moved rostral and caudal to determine the site with the maximal incoming volley. The dura was then removed from the dorsal surface at this region, and this site was used for recordings. A glass microelectrode filled with potassium acetate was inserted in the spinal cord using a motorized micro-drive, and field potentials were recorded at depths corresponding to laminae 1 and 2 using an Axoclamp 2b amplifier (Axon Instruments). These measurements were made at locations in the lateral gray matter indicated in Fig 7C as these regions have been shown to show maximal field potentials (Schouenborg, 1984).

Both the incoming volley and field potentials were further amplified and filtered using Digitimer (UK) amplifiers, digitized using a Power 1401 digital to analogue converter (CED, UK), recorded, and analyzed using Spike 2 software (CED, UK). All measurements were made from averages of at least 10 successive stimulations (performed in Spike 2). The incoming volley was measured as the peak-to-peak amplitude from the first 3 inflections of the cord dorsum potential as illustrated on Fig 7C and D. Baseline recordings were initially performed, followed by application of the peptide to the surface of the spinal cord, and the recordings performed again 20-30 min later in the same location. In 3 mice, the spinal cord was washed immediately after the second measurements were made and recording of the incoming volley repeated after 1 h and 2 h.

## Antibodies

Western blots, fixed primary hippocampal neurons, HEK 93 cells, and tissue sections were analyzed with primary antibodies directed against the following: GFAP (rabbit polyclonal, Dako Cat. No. Z0334; dilution 1:1,000 for immunohistochemistry), NeuN (mouse monoclonal, Millipore Cat. No. MAB377; dilution 1:100 for immunohistochemistry), PICK1 (rabbit polyclonal JH2906 kindly provided from Richard Huganir's laboratory, 5 µg for immunoprecipitation) and PICK1 (mouse monoclonal clone 2G10 custom generated; dilution 1:500 for Western blot), GFP (rabbit polyclonal, Abcam Cat. No. Ab290; dilution 1:500 for immunocytochemistry and Western blot), GluA2 (mouse monoclonal, Millipore Cat. No. MAB397; dilution 1:100 for immunohistochemistry and 1:1,000 for Western blot), pS880 GluA2 (mouse monoclonal, Millipore Cat. No. MABN103; dilution 1: 500 for Western blot), GluA1 (rabbit monoclonal, Millipore Cat. No. 04-855; dilution 1:500 for Western blot), Pan-Cadherin (rabbit polyclonal, Cell Signaling Cat. No. #4068; dilution 1:500 for Western blot), β-actin

### The paper explained

**Problem**

Neuropathic pain affects 7–10% of the world population with higher prevalence in women and elderly. Current medication, including anti-epileptics, antidepressants, and opioids, provides only partial pain relief and comes with considerable side effects. Consequently, there is an urgent need for better treatment. The causes for development of neuropathic pain include diabetes mellitus, treatment with chemotherapeutics, and herpes zoster, which all cause damage to the peripheral nervous system leading to increased central pain perception. This central sensitization involves insertion of excess glutamate receptors in the dorsal horn synapse, and direct modulation of the glutamatergic transmission by glutamate receptor antagonist to obtain better efficacy has been pursued, but clinical development has been discontinued due to adverse side effects.

**Results**

This paper demonstrates a refinement of this approach by targeting the insertion of excess glutamate receptors in the hyper-sensitized condition rather than blocking the receptors themselves. This was achieved by development of a peptide that can inhibit the scaffold protein PICK1, which has previously been evoked as a molecular target in pain treatment because it sustains insertion of a subtype of glutamate receptors. Injection of the peptide in the spinal cord effectively relief pain in an animal model of neuropathy, whereas injection in the paw of the injured hind limb was ineffective, suggesting a central mechanism of action. This was supported by electrophysiological *in vivo* measurement in injured animals, which confirmed that the peptide reduced spinal cord transmission.

**Impact**

The approach presented here evokes a novel mechanism of action for pain relief and potentially provides a potent tool for interfering with excessive glutamate receptor transmission in pain. Importantly, since the peptide only affect disease-related receptor insertion, we do not anticipate major impact on basal neurotransmission. Consequently, with the proper formulation to enable appropriate administration, the peptide is a strong candidate for extended pre-clinical studies leading to future clinical trials for neuropathic pain following peripheral nerve damage.

HRP-conjugated (mouse monoclonal, Sigma Cat. No A3854.; dilution 1:10,000 for Western blot), and IgG-negative controls (normal mouse, Santa Cruz Cat. No. SC-2025, 5 µg; rabbit polyclonal, Cell Signaling 27295, 5 µg for Western blot).

As secondary antibodies, we used Alexa Fluor 488-conjugated goat anti-rabbit IgG (Thermo Fisher scientific, Cat. No. A-11034; dilution 1:500 for immunohistochemistry and immunocytochemistry), Alexa Fluor 647-conjugated goat anti-mouse IgG (Thermo Fisher scientific, Cat. No. A-21236; dilution 1:500 for immunohistochemistry and immunocytochemistry), Alexa Fluor 488-conjugated goat anti-mouse IgG (Thermo Fisher scientific, Cat. No. A-11029; dilution 1:500 for immunocytochemistry), and (HRP)-conjugated secondary antibodies for Western blotting (anti-mouse/anti-rabbit, Abcam Cat. No. ab99632/ab99697; dilution 1:10,000).

## Statistical analysis

Statistical analyses were performed as indicated specifically for individual experiments. Significance level was set to $P < 0.05$. GraphPad

Prism was used for data analyses. For pharmacological, imaging, and biochemical experiments, sample size was chosen according to common practice with at least 3 independent determinations. More replicates were added in several of the biochemical experiments upon reviewer request. For SNI experiments, mice were randomly assigned to individual experimental groups and von Frey testing was performed blinded toward the nature of injected substance and by the same female researcher to avoid person-to-person variation and gender bias. A pre-specified group size of 8 animals was chosen based on previous experience with pharmacology in the SNI model. In 6B and H, animals were taken down prior to experiment reducing the measured n, but in both cases significance was not compromised. Exception is Fig EV4D, where only 4 animals were used for the positive control gabapentin because of its well-known effect size.

## Data availability

This study includes no data deposited in external repositories.

**Expanded View** for this article is available online.

## Acknowledgements

We would like to thank Signe Egstrand Andersen and Mie Vindahl Andersen for help with peptide synthesis of the truncated DAT peptides as well as Nabeela Khadim for technical assistance. We acknowledge the Core Facility for Integrated Microscopy, Faculty of Health and Medical Sciences, University of Copenhagen. We gratefully acknowledge SAXS beam time at the P12 beamline at the PETRAIII at DESY, Germany, along with help from beamline scientist Dr. Haydn D. T. Mertens and co-funding of the beamtime travels from Danscatt. LA and SRM are co-funded by the Lundbeck Foundation Brainstruc project. Danish Council for Independent Research, Technology and Production Sciences (A.B.); Danish Council for Independent Research, Medical Sciences (K.L.M); NOVO Nordisk Foundation, exploratory pre-seed (K.L.M.).

## Author contributions

Peptides were designed by AB, KS, MRa, and KLM. They were synthesized by AB, MS, and KBN, and plasma stability was assessed by CROB. Binding experiments by FP was done and analyzed by NRC, GNH, and MRa. Binding on SCMSs and FPLC was performed and analyses by NRC. Imaging of shRNA experiments was done by IAJ. All other imaging was performed by MDL with hippocampal cultures prepared by SEP. Co-IP, phosphorylation, surface biotinylation, and biotin pull-down were performed by MBL and ABH. Design, recording, and analysis of *in vivo* electrophysiology were done by DBJ and CFM and SNI operations for this done by GNH. *In vivo* behavioral experiments were designed by CBV, MDL, and KLM and carried out and analyzed by MRi, who also performed SNI operations. NMR studies were carried out by SE, NRC, and KT helped analyzing the data. SAXS experiments were carried out by NRC and GNH assisted by SRM, and LA was involved in data analyses and beamtime allocation. ATS, UG, and KLM designed the overall layout of experiments, and MDL, NRC, UG, and KLM wrote the manuscript.

## Conflict of interest

A priority founding patent application (P4997EP00; inhibitors of PICK1 and uses thereof) for Tat-P$_4$-(C5)$_2$ has been filed (KLM, UG, KST, AB, NRC).

## For more information

(i)    http://in.ku.dk/research/Madsen-lab
(ii)   https://in.ku.dk/research/claire-meehan/
(iii)  https://biopharmaceuticals.ku.dk/research/stroemgaard-lab/
(iv)   https://www.iasp-pain.org

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
