## [Review Process File · EMBO Molecular Medicine]

A high-affinity, bivalent PDZ domain inhibitor complexes PICK1 to alleviate neuropathic pain

Nikolaj R. Christensen, Marta De Luca, Michael B. Lever, Mette Richner, Astrid B. Hansen, Gith Noes-Holt, Kathrine L. Jensen, Mette Rathje, Dennis Bo Jensen, Simon Erlendsson, Christian R. O. Bartling, Ina Ammendrup-Johnsen, Sofie E. Pedersen, Michèle Schönauer, Klaus B. Nissen, Søren R. Midtgaard, Kaare Teilum, Lise Arleth, Andreas T. Sørensen, Anders Bach, Kristian Strømgaard, Claire F. Meehan, Christian B. Vægter, Ulrik Gether and Kenneth L. Madsen

Review timeline:

Submission date:	5th Aug 2019
Editorial Decision:	17th Sep 2019
Authors' correspondence :	11th Oct 2019
Editorial Decision:	21st Oct 2019
Authors' correspondence :	4th Nov 2019
Editorial correspondence:	4th Nov 2019
Revision received:	12th Mar 2020
Editorial Decision:	24th Mar 2020
Revision received:	1st Apr 2020
Accepted:	7th Apr 2020

Editor: Celine Carret

Transaction Report:

1st Editorial Decision
2019

17th Sep

Thank you for the submission of your research manuscript to our editorial office. We have now received the enclosed reports on it. Although the reviewers found interest in the question, they felt that the conclusions were not fully supported, and considerable additional work would be necessary. Given this, I am sorry to say that we will not be able to consider the paper further at EMBO Molecular Medicine.

I do want to emphasize that this decision is not intended to imply a lack of interest on our part in either your work in particular or this field in general, and we hope that you will continue to consider EMBO Molecular Medicine for other submissions in the future when it seems appropriate.

***** Reviewer's comments *****

Referee #1 (Comments on Novelty/Model System for Author):

See below

Referee #1 (Remarks for Author):

The role of adapter protein Pick1, which binds PKC and membrane proteins, in nociception at neuropathic pain conditions was explored in the presented study.

Main points:

1. In abstract: "actively disrupts PICK1-receptor complexes". What receptor? Any receptor? Only AMPA receptor? This should be specified in the abstract.
2. NMDA antagonists did not "failed during pre-clinical development". They are often used as positive controls for pain reduction at neuropathic pain models.
3. Why TAT-P4-(C5)₂ peptide is more effective than direct AMPA and NMDA antagonists? Literature and presented results do not support this statement.
4. "PICK1 has been shown to be implicated in central sensitization of neuropathic pain" The cited article shows only effect of Pick ablation on neuropathic pain, they do not show changes in central sensitization, since (1) global KO was used and (2) spinal cord injected antisense can easily reach DRG, which is peripheral tissue.
5. Returning to point 1; Pick1 is expressed in neuronal and non-neuronal cells (please see mouse brain.org). Does it modulate other channels, such as sodium channels and voltage-gated Ca²⁺ channels? Does it modulate receptors in non-neuronal cells? Non-neuronal cell neuron communication?
6. Is TAT-P4-(C5)₂ peptide specific for Pick1?
7. Recording from slices of spinal cord must be performed to show effect of peptide on neurotransmission in spinal cord.
8. The peptide should also be applied peripherally to clarify its effect on DRG fiber excitability using skin-nerve approach. Behavioral experiments with peripheral peptide application should be performed as well.
9. Why 2g von Frey filaments have not been used? Baseline of 1.1-1.2g is very low for C57BL mice.
10. How acute and chronic phases of neuropathic model are identified? Is this subjective perception, or there are some criteria? It is better to use "initiation" and "maintenance" stages of neuropathic pain.
11. Figure 5h is interpreted as "inhibition of neuropathic pain at chronic stage". Why effect is transient, and more pronounced at 28d compare to 14d.

Overall, this study has several critical drawbacks related to (1) novelty, (2) absence of mechanistic studies on neurotransmission in spinal cord and sensory fiber excitability, and (3) over interpretation of data on inhibition of neuropathic pain maintenance by the peptide.

Referee #2 (Remarks for Author):

In this manuscript, De Luca et al. developed a highly potent, cell penetrable, PEG-mediated dimeric PICK1 PDZ domain binder dubbed TAT-P4-(C5)₂. The TAT-P4-(C5)₂ peptide has an apparent PICK1 binding affinity of ~2 nM. The TAT-P4-(C5)₂ peptide can effectively displace PICK1 from its membrane association with target proteins both in heterologous cells and in neurons. Importantly, the TAT-P4-(C5)₂ peptide appears to be able to modulate PICK1-mediated AMPAR synaptic trafficking both in cultured hippocampal neurons and in spinal cord of rodents likely by disrupting PICK1/AMPAR interaction. With these findings in hands, the authors went on the show that the TAT-P4-(C5)₂ peptide can effectively alleviate neuropathic pain sensation of mice. This is a very extensive study from discovery and optimization of the potent PICK1 PDZ binder, the binding mechanism of the peptide to PICK1, all the way to testing physiological function of the developed peptide. The potential value of the developed PICK1 inhibitor in pain management is obvious. The study is built on the past extensive experiences of this collaborative team in targeting PDZ scaffold proteins for modulating physiological functions (such as pain sensation) of the nervous systems. Overall, this is an exciting study supported by a large array of solid experimental data. This reviewer supports publication of the work if the authors can consider a few points below:

1. A key point will be the specificity of the TAT-P4-(C5)₂ peptide. As shown in Fig. S11a-c, TMR-TAT-P4-(C5)₂ by itself can form puncta-like structures in PICK1 negative cells, implying that the peptide may bind to other proteins on plasma membranes. It will be nice if the authors can test binding of the TAT-P4-(C5)₂ peptide to several AMPAR-related PDZ scaffold proteins such as PSD-95.
2. The IP data shown in Fig. 3c&d and Fig. S12 are not very convincing. For example, the bands for GluA2 (or PICK1) in the experimental group is not obviously visible in both "+" and "-" TAT-P4-(C5)₂ peptide lanes in Fig. 3C (or Fig. S12a). This will need to be improved.

3. The SAX data provide useful information in understanding possible conformation changes of PICK1 induced by the binding of the TAT-P4-(C5)₂ peptide. However, the authors should avoid over-fitting the SAX data. A single merged SAX curve fitted by a combination of many binding models does not really mean too much. The authors are suggested to tone down the interpretation of the model derived by the SAX data.

4. It appears that the TAT sequence significantly enhances the PICK1 binding affinity of the C5 peptide (and the P4-(C5)₂ peptide). I suppose that this is a serendipitous finding, which is interesting by itself. It would be nice if some insights can be provided as to why the TAT sequence can enhance the binding of the C5 peptide to PICK1.

Referee #3 (Comments on Novelty/Model System for Author):

I am scoring this manuscript high on novelty and impact because of its potential, should the inconsistencies identified in my review be resolved without undermining the story. Such changes would also increase the clarity and general interest. The technical quality of the work is impressive in some areas, but the presentation is difficult to follow.

Referee #3 (Remarks for Author):

The authors present the development of membrane-permeable, bivalent inhibitors of the PICK1 PDZ domain that exhibit remarkable affinity for their target. More importantly, these bivalent peptides appear able to disrupt PICK1:GluA2 interactions and thus affect GluA2 phosphorylation and internalization. Consistent with these effects, the peptide demonstrates some efficacy in animal models, including a model of neuropathic pain.

Fundamentally, there is lingering question of how affinity is attributed to bivalence. Supplemental Fig. 2a shows that a 5FAM2-P4-(C5)₂ sequence has low nanomolar affinity (unspecified), and the legend asserts that "bivalence increase<s> binding strength compared to monomeric OrG-DAT C11 (dashed line) but the TAT sequence does not further improve binding in contrast to the effect on the monomeric peptide (TAT DAT C5)." I use 5FAM2 as the prefix, because two fluorophores are attached. However, bivalence without fluorophores (and the attachment lysine at P-5) does not have nearly the same effect. P4-(C5)₂ has a reported K_i of 98 nM (p. 6), which is only ~15-fold stronger than C5. The affinity of the 5FAM2-P4-(C5)₂ peptide was not listed in the figure or legend, but appears to be quite strong, suggesting an effect of the fluors (and/or linking lysine) that is not discussed. Furthermore, this raises questions as to whether the bivalency of the ligand is really the source of the remarkable affinity. Fig. 1f also misrepresents the effects of bivalency: TAT-C5 is much closer in apparent affinity to TAT-P4-(C5)₂ (10-fold) than to C5 (100-fold).

Of course, it's possible (but difficult to discern from the description), that the K_i values shown in Fig. 1b were determined using Org-C11 as the tracer. Perhaps the weaker affinity of the tracer underestimates the affinity of the bivalent P4 complex in the same way that it appears to underestimate the affinity of the other TAT-labeled peptides. In Supplemental Figure 3, the K_i values reported for both TAT-labeled peptides are much weaker (ca. 100 nM) than reported elsewhere, using high-affinity tracers. This may reflect the limitations of weaker reporter affinities, but should be clarified and/or addressed.

In the displacement of PICK1 pre-bound to SCSMs, even the bivalent ligand shows efficacy only at concentrations above 1 μ M, which are therapeutically challenging. Also, in the description of the result, the authors suggest that this is because the bivalent complex "increased the macroscopic off-rate of bivalently bound PICK1." Is this clear? Could it not be that the bivalent competitor is simply more effective at preventing rebinding when one of the two PICK1 PDZ domains release C24? If not, what possible mechanism would enable the peptide to affect the PDZ off-rate? The authors suggest that pre-formation of a dimer target might provide a basis for the effect, but the inhibition of dimeric PICK1 rebinding takes place at a single PDZ domain, so how does the availability of a (presumably PICK1-occupied) second site in the vicinity provide a stronger advantage to the bivalent form?

The SAXS data are also somewhat confusing. The bivalent peptide is proposed to stabilize the

dimer-of-dimers assembly. Yet the pair-distribution functions in the presence of peptides show much smaller D_{max} values (Fig. 2h) than do the corresponding pair-distributions functions in the absence of peptides (Suppl. Fig. 7) at all concentrations. How is this consistent with the model?

In terms of the co-immunoprecipitation data, why do TAT-C5 and TAT-P4-(C5)₂ show similar ability to disrupt PICK1:GluA2 interaction (Fig. 3c,d) and block S880 phosphorylation (Fig. 3e,f), when the preceding data had clearly shown the superiority of the bivalent complex? In comparing the effects within the spinal tract, there is a concern that a single (uninhibited) value in Fig. 3c accounts for the difference in significance between the bivalent and monovalent constructs. While recognizing the complexity of such experiments, it is important to test the robustness of the difference. The same caveat applies to S880 phosphorylation (Fig. 4d). Either some additional experiments could be performed or the conclusions could be softened.

Is the dimer of dimers "non-native" as asserted a few times? It appears to form in a concentration-dependent manner in the absence of peptide, and the local concentration of PICK1 in a scaffolding environment may be relatively high. If so, the partnering may, in fact, be physiologically relevant, which would account better for the improved activity of the bivalent vs. monovalent construct in more physiological models.

Additional points:

- why did the authors extensively modify the N-terminus of the C4 peptide, but not the much higher affinity C5 peptide? Fixed position of COOH group means that the N-terminus would be placed differently within the pocket, so no reason to assume that failure at C4 predicts failure at C5.
- need more details on binding experiments, e.g., tracer affinities and concentrations in Supp. Table 2, also for the tracers shown in
- where are the affinities for the tracer peptides in Fig. 1d reported?
- Also - are these measurements actually fluorescence polarization or anisotropy? If FP, how is the concentration dependence of mixed populations fit, since FP values do not combine in a linear fashion?
- Is the curve in Fig. 1d for TMR-P4-(C5)₂ the same as in Supp. Fig. 2b, bottom panel? If so, why is it repeated?
- the role of Triton concentration in the binding experiments is neglected in the manuscript. Does the presence or absence of micelles affect TAT solution behavior? Otherwise, why are the binding experiments performed at paired Triton concentrations?

Typos/presentation:

- p. 7, line 4 refers to Fig. 1c for details of labeling, but those are shown in Supp. Fig. 2 (top panels), except for C5 itself.
- p. 7, line 7: "Table 2" should presumably refer to "Supplementary Table 2."
- p. 7 TMR and TAMRA are used inconsistently.
- p. 7 2nd para "5FAM-TAT-(C5)" but "TAT-C5" -- why parentheses in one case, but not the other? They appear superfluous. (Same question for Fig. 1 panels 1h and 1k.)
- p. 6ff K_i and $K_{i,app}$ are used inconsistently.
- p. 9, line 17: Supplemental Figure 6? (not 5)
- p. 16, line 6: "five residues... is"

- p. 19 "E. coli cultures... was" (verb agreement; species in italics; E. spelled out)
- p. 21 "GripTite cells ... was grown"

- Supp Table 2 appears to use a different nomenclature than the text/figure legends, e.g., TAT11-di-DATC5 vs. TAT11-P4-(C5)2. Or if these are different elements, please describe. The Table also refers to OrgDATC13 whereas the text refers to OrgC11. The legend for Supplemental Fig. 2 refers to this sequence as OrG-DAT C11 (note capital G - lower case in text, but also upper case in Supp. Fig. 1 legend). The legend refers to "TAT DAT C5," which appears to be the same as "TAT C5" described in the manuscript.

- Supp. Fig. 2 legend: "bivalence increase<s>"

- Supp. Fig. 2 legend: should the "Figure 1c" reference be in parentheses at the end?

- Supp. Fig. 3 and legend: why is "TAT" now referred to as "TAT11"? Or are they different? And what happened to P4 in the TAT11-(C5)2 construct in panels 3c and 3f? Is this different? In the legend P4 is still mentioned. What are T1 and T2 in 3c and 3f? Presumably tracers, but not otherwise described. Is T2 aka "fluorescently labeled DAT C11" the same as OrgC11 (p. 6) aka OrG-DAT C11 (Supp Fig. 2 legend)?

- Supp. Fig. 3 legend: 20 nm concentration.

- Supp. Fig. 5 shows the titrations of C5 and P4-(C5)2 displacing pre-bound PICK1 from SCSMs. The titration for C5 appears to show a clear decrease from 1000 nM to 10,000 nM, but the summary curve (Fig. 1k) shows almost no inhibition at 10,000 nM. This could reflect the vagaries of a single representative experiment, but it would be good to show a representative experiment that accounts for/is consistent with the trend shown in Fig. 1k.

- Supp. Fig. 8: what is TPD5? TAT-P4-(C5)2 (D for DAT?)? Why another nomenclature? Or is this different?

- p. 21, line 1: "strenght"

- Fig. 1 legend: panel (c) is not described; description for panel (c) matches panel (d) etc. In the text, the references are to the correct panels.

- Fig. 1d and 1e; why do the legends describe TAT-C5 but TAT11-P4-(C5)2?

- Fig. 1g: "cells expressing... is grown"

- Fig. 1k: "pre-complexred"

- Fig. 2 legend: the description of panel c "abs280 nm" and "abs544 nm" is confusing. The latter is not the PICK1 profile as described in the sentence, but rather the peptide. Also, the nm should be in the subscript. Finally, the axis label says 546 nm. Which is it? And why is Tat now lower-case in the panels, but not the legend?

- Fig 3a: why isn't the panel that contains TMR-C5 (bottom middle) labelled? In the legend, it says "all 20microM" for concentration, but in parentheses after TMR-TAT-C5 it says 5 microM. What concentration(s) is/are used? Fig. 3b says that a 5 nM concentration is used. Is it truly 1000-fold diluted compared to 3a (e.g., to facilitate co-localization)?

- Fig. 3g and 3h are not referred to in the results.

- Fig. 3 title: "TAT-P4-(C5)2 is ... and compromise<s>..."

- Fig. 4 legend: two references to (f), no label for 4g, which is referred to in the text.

Thank you for taking the time to review our manuscript entitled 'A high-affinity PDZ domain inhibitor bivalently complexes PICK1 to alleviate neuropathic pain'. Indeed, we find the points raised by the referees very relevant and agree that addressing them will significantly improve the manuscript. Also, I am very confident that we will be able to provide a point-to-point response to address the criticism raised within a reasonable time-frame. Further, we are pleased that two of the referees appreciate the results in the manuscript, finding them 'exciting' and 'scoring it high on novelty and impact', and that they support publication in EMBO Molecular Medicine if the concerns are appropriately addressed.

It is therefore with much disappointment we see that the editorial decision is to not grant a resubmission to EMBO Molecular Medicine. It seems clear that the main reason for this rejection is the response by Referee 1, who concludes that there are "several critical drawbacks related to (1) novelty, (2) absence of mechanistic studies on neurotransmission in spinal cord and sensory fiber excitability, and (3) over interpretation of data on inhibition of neuropathic pain maintenance by the peptide".

However, these conclusions do not only differ substantially from the conclusions made by the two other reviewers but also, they are not logically connected to the specific criticisms (i.e. "Main Points") raised by the referee. Moreover, these "Main Points" by themselves in several cases are taken out of context, miss/overlook/disregard experiments/results and/or are de facto incorrect.

To summarize, we have the following overall objections, which are specified/detailed in the point-to-point response given below.

1. The point of novelty, which by referee 2 and 3 is clearly identified as the development of a high affinity and efficacious peptide inhibitor of the pain target PICK1 is neglected by referee 1 in the review.
2. The referee takes statements out of context and quotes the manuscript incorrectly (see point 2 and 3)
3. The referee implies wrong use of our references on a role of PICK1 in central sensitization. This is de facto incorrect and even stated at the level of the abstracts (see point 4 below)
4. The referee has missed/overlooked/disregarded several findings that are presented in the paper (see point 3, 5 and 7).

From a scientific perspective, the referee seems preoccupied with a wish to expand the mechanism for PICK1 action beyond the central modulation of AMPA receptors trafficking into other receptor types, peripheral neurons and non-neuronal cells. We agree that this is potentially interesting; however, as should be clear from the manuscript, all literature on PICK1 in pain (ref26-29) revolves around AMPA receptor modulation and this is directly supported by our data. Indeed, PICK1 is the target and we describe here a detailed mechanism for efficacious inhibition representing a novel means to relieve neuropathic pain. Thus, in agreement with the conclusions by Referees 2 and 3, the manuscript provides both novelty and mechanism without any overinterpretation of the data. Accordingly, we would argue that our manuscript is well suited for EMBO Molecular Medicine, and we find it questionable that our manuscript is rejected based on Referee 1's report.

We therefore kindly ask that the decision is reconsidered and ask for permission to resubmit a revised manuscript to EMBO Molecular Medicine with a point-to-point address of all the concerns raised.

Thank you for your patience while we were reconsidering our decision on your reviewed article submitted to EMBO Molecular Medicine.

We have now extensively reviewed the referees' comments in light of your appeal letter and discussed among us. We also have sought further advice from the referees and below are their recommendations. Please note that while we will allow revision of the paper, this sounds like an extensive and time consuming endeavor. As the 3 referees agree on a long list of revision items, we will consider those as mandatory for publication. The revision is a "borderline" invited revision, which means that if you feel that this is too much, please do let us know and we will withdraw the article for you.

Referees' recommendations:

- 1) "in this case, the extent of editing/new experimentation required is so large that it's hard to be confident that the manuscript would warrant publication once all the i's are dotted. What if the missing data, once added, undermine the general interest of the story? Would it be possible to offer a chance at revision, but note that caveat?"
- 2) "the authors should build rational in the way the Reviewer #3 presented. Additionally, they could discuss benefits of this new strategy. However, these benefits should be discussed from stand-point of side effects, tolerance and efficacy. Hence, they should be somehow experimentally supported."
- 3) "the mechanism is not very clear. [...] It will be critical that the pain physiology data presented must be clean. If the authors can provide additional supporting data pointing to possible mechanistic aspect, the manuscript may deserve a consideration for publication."

Should you agree to revise your article for EMBO Molecular Medicine, please be reminded that the new data may (or may not) be sent again to all referees (at the editor's discretion); ref. 2 & 3 comments must be addressed experimentally when needed; from ref. 1's comments, check whether Pick1 blockage alters neurotransmission in spinal cord and sensory fiber, show peptide specificity and overall provide well controlled and detailed data.

We do realize that the criticisms are many. While we do not request detailed mechanisms, your basic concept must be convincingly proven.

I should remind you that it is EMBO Molecular Medicine policy to allow a single round of revision only and that, therefore, acceptance or rejection of the manuscript will depend on the completeness of your responses included in the next, final version of the manuscript. For this reason, and to save you from any frustrations in the end I would strongly advise against returning an incomplete revision and would also understand your decision if you chose to rather seek rapid publication elsewhere at this stage.

Thank you for considering our appeal and re-evaluating the review. We would like to meet the challenge and submit a revised manuscript!

While the reviews are overall clear and unambiguous, we have two questions to specify the requests by referee 1:

7. Recording from slices of spinal cord must be performed to show effect of peptide on neurotransmission in spinal cord

We have a strong wish to test whether the peptide modulate transmission in the spinal cord. Importantly, however, we do not anticipate the peptide to modulate the transmission under normal conditions as we don't see any significant change in surface expression of AMPARs (Fig4). Changes in surface receptor levels are only observed after SNI (Fig5). The SNI model is done in

mice from postnatal week 8 and onwards and in our case from week10. Electrophysiological recordings from slices of mice past p20 is challenging to the level where, to my knowledge, only a selected few groups can do this. Would an ex vivo spinal cord LTP-like experiment like in 'Synaptic amplifier of inflammatory pain in the spinal dorsal horn.' Science. 2006 Jun 16;312(5780):1659-62 Ikeda H1, Stark J, Fischer H, Wagner M, Drdla R, Jäger T, Sandkühler J. address the reviewers point?

'8. The peptide should also be applied peripherally to clarify its effect on DRG fiber excitability using skin-nerve approach. Behavioral experiments with peripheral peptide application should be performed as well.'

What is meant by peripheral peptide application? The present manuscript showed i.p administration in the SNI model giving a reduced relief of allodynia compared to i.t. injection. We now also have data from subcutaneous administration. Would that answer the question or is the request rather a topical administration to the affected paw? 'Peripheral peptide application' is somewhat ambiguous.

Editor's correspondence

4th Nov 2019

Thank you for your e-mail and for contacting us regarding the expected revision of your article.

I have contacted the referee 1 especially regarding the 2nd point you mentioned as I couldn't answer to that myself (I'm not an expert in pain research).

Please find her/his answers below:

" 1. Authors wrote: "we don't see any significant change in surface expression of AMPARs (Fig4)."  Experiments have to be performed for control and experimental conditions. [...] AMPA could be sensitized without surface expression change. For example, phosphorylation could occur.

2. Authors wrote: "Electrophysiological recordings from slices of mice past p20 is challenging to the level where, to my knowledge, only a selected few groups can do this."  It is relatively challenging [...]. Neurotransmission efficiency is checked only one way. Please read works by H-L Pan group from MD Anderson, Houston.

3. Authors wrote: "What is meant by peripheral peptide application?"  It is standard experiments when peripheral versus central effect of drug is evaluated. [...] For example, the authors could look classical works on peripheral vs central effect of opioids in suppressing of pain. "

To the initial point 7 of the referee therefore I would answer that it would be great to try to perform the experiment in vivo at P20. If however, this feels too troublesome, an ex vivo approach will be warranted.

To the initial point 8 of the referee, I am afraid that I can't answer to that but hope that you will find information following the referee's recommendations.

2nd revision – authors' response

12th Mar 2020

Referee #1 (Remarks for Author):

Main points:

1. In abstract: "actively disrupts PICK1-receptor complexes". What receptor? Any receptor? Only AMPA receptor? This should be specified in the abstract.

*The section in the **abstract** reads:*

'Tat-P4-(C5)₂, but not a monomeric variant (Tat-C5), actively disrupts PICK1-receptor complexes on supported cell membrane sheets'

And refers specifically to the SCMS experiments, which take advantage of the synthetic construct TacDATC24, but likely applies in general to interaction partners in the membrane including the AMPAR. We suggest changing the wording the read:

'Tat-P4-(C5)₂, but not a monomeric variant (Tat-C5), actively disrupts PICK1 interaction with membrane proteins on supported cell membrane sheets'

2. NMDA antagonists did not "failed during pre-clinical development". They are often used as positive controls for pain reduction at neuropathic pain models.

The sentence read "major efforts have been directed towards developing compounds targeting the ionotropic NMDA (N-methyl-D-aspartate) and AMPA (α-amino-3-hydroxy-5-methyl-4-isoxazolepropionic acid)-type glutamate receptors, but such compounds have generally failed during pre-clinical development or in clinical trials due to lack of efficacy or as a result of unacceptable side effects" This sentence does not specifically relate to pain, but to diseases involving glutamatergic dysfunction. It is true that NMDA receptor antagonists work in animal models with peripheral nerve injury but clinical application is seriously compromised by a very narrow therapeutic index. AMPA receptor antagonists have been less efficacious with the exception of perampanel, which however displays very severe side-effects. Nonetheless, to avoid confusion, we have changed the text to specify only clinical trials and reads:

'Indeed, major efforts have been directed towards developing compounds targeting the ionotropic NMDA (N-methyl-D-aspartate) and AMPA (α-amino-3-hydroxy-5-methyl-4-isoxazolepropionic acid)-type glutamate receptors, but such compounds have generally failed during clinical development due to lack of efficacy or as a result of unacceptable side effects (Tymianski, 2014).'

3. Why TAT-P4-(C5)₂ peptide is more effective than direct AMPA and NMDA antagonists? Literature and presented results do not support this statement.

We do not claim that TAT-P4-(C5)₂ is more effective than AMPA receptor and NMDA receptor antagonist. What we do is to promote the idea that it is possible "to target the synaptic scaffold proteins that orchestrate synaptic signaling complexes and dynamically regulate the surface expression and ion-conductance of the ionotropic glutamate receptors in the postsynaptic density" The interesting rationale is that this might produce fewer side-effects and a better therapeutic window, simply because one will never fully block glutamatergic transmission but only the subset of receptors under trafficking e.g. during plasticity. Indeed, we have experimental evidence to support this hypothesis in the biotinylation studies in figure 5 and 6, which show that TAT-P4-(C5)₂ is more efficacious in reducing GluA1 and 2 surface expression after injury

4. "PICK1 has been shown to be implicated in central sensitization of neuropathic pain" The cited article shows only effect of Pick ablation on neuropathic pain, they do not show changes in central sensitization, since (1) global KO was used and (2) spinal cord injected antisense can easily reach DRG, which is peripheral tissue.

We cite four papers in this context:

Atianjoh, F.E. et al. Spinal cord protein interacting with C kinase 1 is required for the maintenance of complete Freund's adjuvant-induced inflammatory pain but not for incision-induced post-operative pain. Pain 151, 226-34 (2010).

Garry, E.M. et al. Specific involvement in neuropathic pain of AMPA receptors and adapter proteins for the GluR2 subunit. Mol. Cell Neurosci. 24, 10-22 (2003).

Wang, W. et al. Preserved acute pain and impaired neuropathic pain in mice lacking protein interacting with C Kinase 1. Mol Pain 7, 11 (2011).

Wang, Z. et al. PICK1 Regulates the Expression and Trafficking of AMPA Receptors I Remifentanyl-Induced Hyperalgesia. Anesth Analg 123, 771-81 (2016)

Garry et al, first demonstrated that PICK1 regulates surface expression and phosphorylation of spinal cord AMPARs in response to the CCI-model of neuropathic pain using i.t. administration of myristoylated peptides. In the peer reviewed abstract, Garry et al state that "that AMPA receptors play a role in the central sensitisation that is thought to underpin chronic pain" and that "We

implicate for the first time a possible role for GRIP, PICK1 and NSF in neuropathic sensitisation from experiments with cell-permeable blocking peptides mimicking their GluR2 interaction motifs". This is the context for the three other references. Atianjoh et al likewise find changes in GluA2 trafficking and phosphorylation in the dorsal horn in the CFA-model of inflammatory pain and claim in the abstract "our findings suggest that spinal PICK1 may participate in the maintenance of persistent inflammatory pain". It is true that none of the approaches are selective to the spinal cord and that both the peptides and antisense can reach DRGs. Consequently, it is not possible to conclude that the role of PICK1 in central sensitization underlies the behavioral response to ablation of PICK1. Nevertheless, the finding that PICK1 is implicated in central sensitization is uncontested by this potential complication to the impact on behavior.

5. Returning to point 1; Pick1 is expressed in neuronal and non-neuronal cells (please see mouse brain.org). Does it modulate other channels, such as sodium channels and voltage-gated Ca²⁺ channels? Does it modulate receptors in non-neuronal cells? Non-neuronal cell neuron communication?

We are fully aware that PICK1 expression is not confined to neurons – we have even reported on on the function of PICK1 in endocrine tissue (Holst, Madsen et al, Plos Bio, 2013). The expression in glia has been somewhat controversial but a putative role in non-neuronal cell-neuron communication pose an interesting question but we find that this is outside the scope of the the present study. Moreover, the peptide displays a surprising neuronal tropism (NeuN) (see figure 5F) and, importantly, we find no overlap with the glial marker GFAP (supplementary figure 15).

Among the 40+ interaction partners reported for PICK1, no voltage gated sodium and calcium channels have been reported, the modulation of ASIC could potentially be relevant to the role of PICK1 in pain and we are currently addressing this question

6. Is TAT-P4-(C5)₂ peptide specific for Pick1?

We have provided several lines of evidence to support the specific of TAT-P4-(C5)₂ peptide Pick1 as detailed below in response to reviewer 2 – point 1. We have screened the peptide sequence against 40+ PDZ domains, ruled out interaction with PSD-95 and outlines outlined the molecular mechanism and specificity of the Tat-sequence interacting with the PICK1 PDZ domain.

7. Recording from slices of spinal cord must be performed to show effect of peptide on neurotransmission in spinal cord.

We agree that this is a central point however slices electrophysiology on mice that are old enough for the SNI model is very challenging.

Instead we have been able to perform the experiment in vivo in decerebrate mice subjected to the SNI procedure. In brief, we find by field recordings that application of TAT-P4-(C5)₂ indeed reduce transmission in layer 1 and 2 of the dorsal horn and that it also reduces the incoming fiber volley in agreement with previous reports on the expression of PICK1 both in layer 1 and 2 neurons in the dorsal horn as well as in DRGs.

Moreover, giving the involvement of PICK1 in expression of Calcium Permeable AMPAR in several tissues in disease we also addressed whether TAT-P4-(C5)₂ could reverse TNF α -induced expression of CP-AMPA in spinal cord slices. This was indeed the case further substantiating effects on spinal cord transmission.

*These finding are presenter together in **new figure 7 and text on page 17-18***

8. The peptide should also be applied peripherally to clarify its effect on DRG fiber excitability using skin-nerve approach.

*We agree that this is an important point and have now performed experiment with intraplantar injection of the same and 10-fold higher dose as in the i.t. experiments without significant alleviation of allodynia, suggesting little peripheral effect of Tat-P4-(C5)₂. This new data is presented on **page 15, last paragraph** and gave rise to **new Appendix FigS11***

‘Intraplantar administration of the same dose did not affect PWT and even a 10-fold higher intraplantar dose did not significantly alleviate allodynia, suggesting little peripheral effect of Tat-P4-(C5)₂ (Appendix Fig S11).’

9. Why 2g von Frey filaments have not been used? Baseline of 1.1-1.2g is very low for C57BL mice.

We disagree that 1.1-1.2g represents a low baseline mechanical threshold for mice. Indeed, this value is in accordance with previous published reports studying adult C57bl/6 mice, including the establishment of the model in mice (e.g. Figs. 2 and 3 in Bourquin AF et al. Pain (2006) May;122(1-2):14.e1-14 Assessment and analysis of mechanical allodynia-like behavior induced by spared nerve injury (SNI) in the mouse. PMID: 16542774 and in e.g. Fig. 3 in Richner M et al., Science Advances (2019) Jun 19;5(6):eaav9946 Sortilin gates neurotensin and BDNF signaling to control peripheral neuropathic pain PMID: 31223654).

Mechanical threshold was assessed by application of manual Semmes-Weinstein monofilaments (Stoelting Co) as described in the methods section. After reaching threshold, it provides no further information to test the 2g filament. Apparent higher threshold values are often seen from groups using the automated von Frey test, e.g. Fig. 2A in Leo et al. Behavioural Brain Research (2008) Jul 19;190(2):233-42 Differences in nociceptive behavioral performance between C57BL/6J, 129S6/SvEv, B6 129 F1 and NMRI mice, PMID: 18420287, and in Fig. 1A in Martinov T et al., Journal of Visualised Experiments (2013) (82): 51212. Measuring Changes in Tactile Sensitivity in the Hind Paw of Mice Using an Electronic von Frey Apparatus, PMID:24378519. Perhaps this is what is referred to?

10. How acute and chronic phases of neuropathic model are identified? Is this subjective perception, or there are some criteria? It is better to use "initiation" and "maintenance" stages of neuropathic pain.

The proposed terminology has been adopted in the legend for new Fig 6 and EV4 and the use of acute and chronic has been omitted in the abstract.

11. Figure 5h is interpreted as "inhibition of neuropathic pain at chronic stage". Why effect is transient, and more pronounced at 28d compare to 14d.

We do not intend to imply that the inhibition by the peptide is chronic but that the pain state in the late phase better mimics chronic pain and, as discussed in 10, we agree to describe this as the maintenance phase. The effect of the peptide is transient at this stage as it is in the early phase of the model. The most likely explanation is that we do not interfere with the aberrant high frequency input from DRGs produced by the injury (although we do reduce amplitude) but rather with the postsynaptic response to this in the dorsal horn secondary neurons. As the effect of the peptide wears off, the maladaptive plasticity is likely reinitiated. We do not know why the effect is more potent at day 28 than at day 14, but this is an attractive aspect of a putative drug

Referee #2 (Remarks for Author):

1. A key point will be **the specificity of the TAT-P4-(C5)₂ peptide**. As shown in Fig. S11a-c, TMR-TAT-P4-(C5)₂ by itself can form puncta-like structures in PICK1 negative cells, implying that the peptide may bind to other proteins on plasma membranes. It will be nice if the authors can test binding of the TAT-P4-(C5)₂ peptide to several AMPAR-related PDZ scaffold proteins such as PSD-95.

We agree that specificity of is important to address to better understand the mechanism of action as well as putative off-target side-effects.

*To this end we now include data suggesting high specificity among a broad range of purified PDZ domains, including PDZ domains from SAP97, SAP102, SHANK and GRIP, for the DAT C5 sequence (**Fig EV1C**) now described on **page 6 second paragraph**:*

'We evaluated the specificity of C5 (EV1C) for a broad selection of PDZ domains (both class I and II) that were previously purified and had known ligands (Stiffler et al, 2007). C5 (10 μ M) (black column) competed for binding to PICK1 as seen by the reduction in normalized mP (milli polarization) compared to no peptide (full line). The C5 peptide primarily bound to the PDZ domain of PICK1, however with notable exceptions for Scribble (Scrb1) PDZ 2/4, Na(+)/H(+) exchange regulatory cofactor (NHERF) PDZ 2/2 and E3 ubiquitin-protein ligase PDZRN3 (Semcap3) PDZ 1/2, which were inhibited to similar level as observed for PICK1 (indicated by dashed line) (EV1C). C5, however, did not compete with binding of either domain in full binding curves, suggesting they were false positives in the screen (Appendix Fig 1).'

and TAT C5 (Figure 1G), described on page 7, bottom of first paragraph:

'To test the specificity, we tested TAT-C5 (10 μ M) in the PDZ domain screen and it did not reduce the normalized mP to the level seen for PICK1 (indicated by dashed line) for any of the other PDZ domains suggesting that the TAT peptide increased specificity (Fig 1F).'

The binding of TAT-P4-(C5)₂ to PSD95 presenting the tandem PDZ12 configuration was also evaluated. Whereas the PSD-95 specific peptide AB125 showed robust binding to PSD-95, TAT-P4-(C5)₂ did not show any binding (Fig EV1F), now described on page 8, bottom of first paragraph:

'Finally, no binding of TAT-P4-(C5)₂ was observed to another bivalent target, PSD95, whereas an analogous dimeric inhibitor with a PSD95 specific sequence showed potent binding (Bach et al, 2009) (Fig EV1F).'

Given the high homology among the MAGUK type PDZ domain proteins including SAP07 and SAP102, we believe that interaction with these proteins is equally low.

As for the puncta-like structures of TMR-TAT-P4-(C5)₂ in PICK1 negative cells, they most likely represent peptide that has not yet escaped endocytic structures upon internalization.

2. The **IP data** shown in Fig. 3c&d and Fig. S12 are not very convincing. For example, the bands for GluA2 (or PICK1) in the experimental group is not obviously visible in both "+" and "-" TAT-P4-(C5)₂ peptide lanes in Fig. 3C (or Fig. S12a). This will need to be improved.

We agree that the bands and Co-IP show were not fully convincing. We have now excluded the old data with weak band from quantification and repeated the Co-IP from hippocampal slices to produce a better quality western blot. To enable us to pool results from different experiments (individual antibody incubations and exposures of the blots) the quantification is now normalized to control within experiments, resulting in a control column without variation. With the addition of the new experiments, TAT-C5 does not affect the PICK1-GluA2 Co-IP, while the effect of TAT-P4-(C5)₂ is still observed. We also increase replicates for spinal cord as detailed in response for reviewer 3 below.

3. The **SAXS data** provide useful information in understanding possible conformation changes of PICK1 induced by the binding of the TAT-P4-(C5)₂ peptide. However, the authors should avoid over-fitting the SAX data. A single merged SAXS curve fitted by a combination of many binding models does not really mean too much. The authors are suggested to tone down the interpretation of the model derived by the SAXS data.

We realize, that in the writing it was not entirely clear, which conclusion was based on the model independent SAXS data and which was based on the EOM modeling. Importantly, the main conclusion regarding the formation of a stable, compact, tetrameric configuration of PICK1 in presence of the TAT-P4-(C5)₂ peptide rely exclusively of the model independent data showing I_o that is concentration independent and a significantly confined D_{max}. To highlight that this is derived from the model independent data we tightened this part of the conclusion bases on the primary data analysis:

'In conclusion, this demonstrates the formation of a stable, compact, tetrameric PICK1 complex by TAT-P4-(C5)₂.'

To acknowledge the ambiguous nature of the EOM analysis, we modified the intro to this part to read 'investigate the structural organization of the complex' rather than 'determine the structural organization of the complex' and now show the EOM analysis as expanded view to tone it down.

Finally, the overall conclusion on the SAXS data is further refined to reflect conclusions drawn from the primary (model independent) data analysis and the EOM analysis respectively:

'Taken together the model independent SAXS analysis show that that TAT-P4-(C5)₂ induces a stable, compact, tetrameric state of PICK1 with the EOM analysis suggesting configurations with the PDZ domains from two individual PICK1 dimers within distances that can be bridged by the bivalent peptide.'

4. It appears that the **TAT sequence significantly enhances the PICK1 binding affinity of the C5 peptide** (and the P4-(C5)₂ peptide). I suppose that this is a serendipitous finding, which is interesting by itself. It would be nice if some insights can be provided as to why the TAT sequence can enhance the binding of the C5 peptide to PICK1.

*Full length PICK1 has not been crystalized and is too for PICK1 NMR studies and the truncated PDZ domain is not stable. Fortunately, however, a construct has been made, with a C-terminal extension binding back in the PDZ domain binding groove that is amenable to NMR studies. We cleaved of the extension of this construct to allow exchange with TAT-C5 and recorded chemical shift changes in the process to see which residues in the PDZ domains might be affected by the TAT sequence. These residues line up in a pattern extending from the binding groove and all the way to the backside of the domain suggesting that the affinity gain is obtained locally within the PDZ domain. This **data is now described on page 7 first paragraph** and gave rise to **new Fig1D-F**. Also, a PDZ specificity screen (**new Fig1G**) suggested that that affinity gain achieved by addition of the TAT sequence was specific for the PICK1 PDZ domain.*

Referee #3 (Comments on Novelty/Model System for Author):

Fundamentally, there is lingering question of **how affinity is attributed to bivalence**. Supplemental Fig. 2a shows that a 5FAM2-P4-(C5)₂ sequence has low nanomolar affinity (unspecified), and the legend asserts that "bivalence increase<s> binding strength compared to monomeric OrG-DAT C11 (dashed line) but the TAT sequence does not further improve binding in contrast to the effect on the monomeric peptide (TAT DAT C5)." I use 5FAM2 as the prefix, because two fluorophores are attached. However, bivalence without fluorophores (and the attachment lysine at P-5) does not have nearly the same effect. P4-(C5)₂ has a reported K_i of 98 nM (p. 6), which is only ~15-fold stronger than C5. The affinity of the 5FAM2-P4-(C5)₂ peptide was not listed in the figure or legend, but appears to be quite strong, suggesting an effect of the fluors (and/or linking lysine) that is not discussed. Furthermore, this raises questions as to whether the bivalency of the ligand is really the source of the remarkable affinity. Fig. 1f also misrepresents the effects of bivalency: TAT-C5 is much closer in apparent affinity to TAT-P4-(C5)₂ (10-fold) than to C5 (100-fold).

We are grateful for the reviewer pointing out these discrepancies in our original presentation of the data. We have now fundamentally rearranged the description of the binding data in the first data section to start off with TAT-C5 and the affinity gain by Tat. We also include NMR data to support interaction of the Tat peptide with the PDZ domain in a pattern extending from the PDZ binding groove an all the way to the back of the PDZ domain. We also have added a screen among 40+ PDZ domains which suggest that this affinity gain is selective for PICK1. We then go on to describe the development of the dimeric approach and the affinity gain achieved by adding the Tat-peptide. We believe this part is now significantly more coherent and thank the reviewer for pointing this out.

This is now reflected also in the abstract, where this section has been changed to read:

‘The affinity gain is obtained partly from the Tat-peptide and partly from the bivalency of the PDZ motif, engaging PDZ domains from two separate PICK1 dimers to form a complex.’

Of course, it's possible (but difficult to discern from the description), that the K_i values shown in Fig. 1b were determined using Org-C11 as the tracer. Perhaps the weaker affinity of the tracer underestimates the affinity of the bivalent P4 complex in the same way that it appears to underestimate the affinity of the other TAT-labeled peptides. In Supplemental Figure 3, the K_i values reported for both TAT-labeled peptides are much weaker (ca. 100 nM) than reported elsewhere, using high-affinity tracers. This may reflect the limitations of weaker reporter affinities, but should be clarified and/or addressed.

*It is indeed true that the competitive FP assay is very sensitive to the affinity of the tracer used. In particular, low affinity tracers will give rise to massive inhibitor depletion in the inhibitor is high affinity (see Appendix Fig 3). This is now highlighted in the text bottom of **second paragraph page 8***

‘Importantly, the peptides also potentially competed with DAT and GluA2 tracers although this was challenging to assess due to inhibitor depletion in the assay (Appendix Fig S3).’

saturation binding curves on the other hand are sensitive to effects of the fluorophore of the peptide. In consequence, the best way to assess the binding strength is by having tracer of competitors of matched affinities which we can achieve in all cases.

In the **displacement of PICK1 pre-bound to SCSMs**, even the bivalent ligand shows efficacy only at concentrations above 1 microM, which are therapeutically challenging.

*This is true and likely we will not reach in vivo concentrations in this range, however, in the experiment the PICK1 pre-binding to the sheet is pretty high or us to detect it in the assay, which is likely to give rise to depletion of the inhibitor thereby shifting the observed efficacy. Likely, PICK1 density at the plasma membrane in neurons is lower, which means that we can probably not use the absolute potency obtained from the experiment, but rather the relative difference between the inhibitory peptides. This is now added to the conclusion of the results on bottom of **page 9***

‘PICK1 pre-binding to the sheet is likely to give rise to depletion of the inhibitors, which means that we can probably not translate the absolute potency obtained from the experiment to a neuronal setting. The relative difference however is striking and suggests a unique advantage of the dimeric ligand in a therapeutic context where the ability to dissociate a preformed complex predictably would be highly advantageous.’

Also, in the description of the result, the authors suggest that this is because the bivalent complex "increased the macroscopic off-rate of bivalently bound PICK1." Is this clear?

Yes, the way the experiment is done is to allow release of PICK1 after washing excess protein away. Since TAT-P4-(C5)2 reduces binding compared to the other peptides it must increase the macroscopic off-rate.

Could it not be that the bivalent competitor is simply more effective at preventing rebinding when one of the two PICK1 PDZ domains release C24? If not, what possible mechanism would enable the peptide to affect the PDZ off-rate?

*We believe that it is indeed an interference with rebinding, but likely related to membrane binding of the amphipathic helix preceeding the BAR domain and the CPC loop in the PDZ domain since we have recently demonstrated that they are key more important for avidity in the assay the multiple PDZ bindings. We offer the following explanation for the finding in the **discussion, Page 21:***

‘We propose that PICK1 may assemble in a tetrameric configuration, similar to our SAXS structure (Fig 3L), upon binding to membrane embedded receptors. In this scenario, peptide

binding (monovalent as well as bivalent) will interfere with PDZ interaction with ligand in the membrane. We recently demonstrated, however, that PICK1 achieve most of the avidity, not only from multiple PDZ interaction at the membrane the avidity, but rather from interaction with the lipid membrane by the amphipathic helix lining the BAR domain {Erlendsson, 2019 #1344; Herlo, 2018 #1281} and likely also the CPC-loop in the PDZ domains {Shi, 2010 #801} (Fig EV5, black dashed circles). Interestingly, according to our SAXS structure (Fig 3L), binding of Tat-P4-(C5)₂ restricts orientation of the two PDZ domains, which may sterically compromise these membrane-interacting motifs and thereby also the rebinding events they govern (Fig EV5, bottom). Since TAT-C5 will not restrict the PDZ orientation (Fig EV5, bottom) this could explain why only Tat-P4-(C5)₂ facilitate the macroscopic off-rate'

We also illustrate the model in new figure EV5

The authors suggest that pre-formation of a dimer target might provide a basis for the effect, but the inhibition of dimeric PICK1 rebinding takes place at a single PDZ domain, so how does the availability of a (presumably PICK1-occupied) second site in the vicinity provide a stronger advantage to the bivalent form?

In the model, we envision tetrameric configuration of PICK1, as suggested also in the review below.

The SAXS data are also somewhat confusing. The bivalent peptide is proposed to stabilize the dimer-of-dimers assembly. Yet the pair-distribution functions in the presence of peptides show much smaller D_{max} values (Fig. 2h) than do the corresponding pair-distributions functions in the absence of peptides (Suppl. Fig. 7) at all concentrations. How is this consistent with the model?

We acknowledge that the suggestion of formation of a dimer-of-dimers in presence of the peptide may at first seem counterintuitive in view of the reduced D_{max} of PICK1. The critical point to appreciate, however, is our previously published finding that PICK1 in absence of peptide forms elongated dimers-of-dimers as well as higher order structures. To make this more transparent, we have now put forward the SAXS data on the WT PICK1 into the primary figure 2 as well as integrated the elongated configuration of PICK1 in absence of peptide in the model. The model now recapitulates the transition from the published complex mixture of dimers and elongated dimers of dimers as well as higher oligomeric configurations in absence of peptide into predominantly tetramers with a lower D_{max} in presence of the peptide consistent with formation of a compact, tetrameric state of PICK1 as described in the response to reviewer 2 above.

In terms of the **co-immunoprecipitation data**, why do TAT-C5 and TAT-P4-(C5)₂ show similar ability to disrupt PICK1:GluA2 interaction (Fig. 3c,d) and block S880 phosphorylation (Fig. 3e,f), when the preceding data had clearly shown the superiority of the bivalent complex? In comparing the effects within the spinal tract, there is a concern that a single (uninhibited) value in Fig. 3c (*we take it this refers to old figure 4c and not 4c?*) accounts for the difference in significance between the bivalent and monovalent constructs. While recognizing the complexity of such experiments, it is important to test the robustness of the difference. The same caveat applies to S880 phosphorylation (Fig. 4d). Either some additional experiments could be performed or the conclusions could be softened.

As described above we have added experiments on the PICK1-GluA2 Co-IP (new Fig 4C and D) and we also added more replicates to the Co-IP in spinal cord (new Fig 5B and C) and the GluA2 S880 (new Fig 5D) upon i.t. injection of peptides. To enable us to pool results from different experiments (individual antibody incubations and exposures of the blots) the quantification is now normalized to control within experiments, resulting in a control column without variation. With the addition of the new experiments, TAT-C5 does not affect the PICK1-GluA2 Co-IP, while the effect of TAT-P4-(C5)₂ is still observed. The previous conclusion on spinal cord is unchanged by additional replicates.

In summary, the Co-IP experiment suggest that TAT-P4-(C5)₂ is indeed more potent at interfering with the PICK1 GluA2 interaction than Tat-C5 in agreement with the in vitro FP and SCMSs data.

The PICK1-PKC interaction, thought to underlie the s880 phosphorylation, however, is more labile to interference and was also affected by Tat-C5. This is in agreement with the fact that it does not benefit from multivalent interactions with PICK1.

Is the **dimer of dimers "non-native"** as asserted a few times? It appears to form in a concentration-dependent manner in the absence of peptide, and the local concentration of PICK1 in a scaffolding environment may be relatively high. If so, the partnering may, in fact, be physiologically relevant, which would account better for the improved activity of the bivalent vs. monovalent construct in more physiological models.

This statement was based on the fact that the configuration of PICK1 obtained in complex with TAT-P₄-(C5)₂ is not a major species in our previously published SAXS data. The species that forms in a concentration dependent manner is more like a head-to-tail arrangement with a slight overlap. We agree, however, that the side-by-side dimer-of-dimer configuration enforced by TAT-P₄-(C5)₂ may be physiologically relevant upon binding to multivalent interaction partners in the membrane, which is now covered in the discussion and a model for this binding shown in Fig EV5. We have abandoned the use of non-native configuration and instead used novel configuration.

Additional points:

- why did the authors extensively modify the N-terminus of the C4 peptide, but not the much higher affinity C5 peptide? Fixed position of COOH group means that the N-terminus would be placed differently within the pocket, so no reason to assume that failure at C4 predicts failure at C5.

This is absolutely true and in hindsight that might have been a better approach to modify DAT C5 more extensively. Our rationale was that since we lost affinity from C5 to C4 we would be able to obtain some affinity in this area, while keeping the resulting molecule small. We have rephrased the sentence describing this experiment to better reflect the rationale:

'Attempts to regain affinity for the DAT C4 peptide by various N-terminal modifications while keeping the resulting molecule relatively small, similar to Bach, et al. 35, were unsuccessful, as were simple additions to the DAT C3 and C5 peptides (supplementary Table 1).'

- need more details on binding experiments, e.g., tracer affinities and concentrations in Supp. Table 2, also for the tracers shown in

this was never a full sentence

- where are the affinities for the tracer peptides in Fig. 1d reported?

Tracer identities are now included in Appendix Table S2

- Also - are these measurements actually fluorescence polarization or anisotropy? If FP, how is the concentration dependence of mixed populations fit, since FP values do not combine in a linear fashion?

While we have used fluorescence polarization for historical reason, we fully acknowledge the point made by the reviewer. Indeed, the term for solving the equation to isolate occupancy becomes very difficult, we argue that the error introduced by assuming linear combination of mP is very limited. To support this argument, we have compared the curves obtained for saturation binding of TMR-TAT-P₄-(C5)₂ (which has the biggest ΔmP in the manuscript) assuming linear combination of mP with that of using the term for anisotropy

As can be seen from the curves the error is indeed minute.

- Is the curve in Fig. 1d for TMR-P4-(C5)2 the same as in Supp. Fig. 2b, bottom panel? If so, why is it repeated?

This whole section has been changed

- the role of Triton concentration in the binding experiments is neglected in the manuscript. Does the presence or absence of micelles affect TAT solution behavior? Otherwise, why are the binding experiments performed at paired Triton concentrations? Clarify this point

The 0.01% Triton, which is below the CMC of 0.015%, is essential in our hands to keep PICK1 in solution. This gave way to the first determination of in-solution binding affinities of the PICK1 PDZ domain (Madsen et al, JBC, 2005) and has been used subsequently for binding studies on PICK1 (Thorsen et al, PNAS 2010 and Erlendsson et al, JBC, 2014). At 0.1% triton the PICK1 BAR domain homodimerization is compromised (described in Karlsen et al, Structure, 2015) and we use this feature in the present paper (suppl Fig.6 – described in p9) to probe the role of the PICK1 homodimerization for the binding of the dimeric peptide. This role of Triton-X 100 is now stated in the legend for Suppl Table 2 for clarity.

Typos/presentation:

-p. 7, line 4 refers to Fig. 1c for details of labeling, but those are shown in Supp. Fig. 2 (top panels), except for C5 itself.

This whole section has been changed

- p. 7, line 7: "Table 2" should presumably refer to "Supplementary Table 2."
corrected

- p. 7 TMR and TAMRA are used inconsistently.

Tetramethylrhodamine is now abbreviated (TMR) on first use and used as TMR throughout

- p. 7 2nd para "5FAM-TAT-(C5)" but "TAT-C5" -- why parentheses in one case, but not the other? They appear superfluous. (Same question for Fig. 1 panels 1h and 1k.)

Tat—(C5) has been changed to Tat-C5 throughout

- p. 6ff Ki and Ki,app are used inconsistently.

We agree, we now use Ki consistently throughout

- p. 9, line 17: Supplemental Figure 6? (not 5)

Corrected

- p. 16, line 6: "five residues... is"

Corrected

- p. 19 "E. coli cultures... was" (verb agreement; species in italics; E. spelled out)

Corrected

- p. 21 "GripTite cells ... was grown"

Corrected

- Supp Table 2 appears to use a different nomenclature than the text/figure legends, e.g., TAT11-di-DATC5 vs. TAT11-P4-(C5)2. Or if these are different elements, please describe. The Table also refers to OrgDATC13 whereas the text refers to OrgC11. The legend for Supplemental Fig. 2 refers to this sequence as OrG-DAT C11 (note capital G - lower case in text, but also upper case in Supp. Fig. 1 legend). The legend refers to "TAT DAT C5," which appears to be the same as "TAT C5" described in the manuscript.

OG-DAT as tracer is indeed C11. This is now corrected in Supp Table 2. Tat in all cases is Tat11 and is now consistently written as just Tat. OrG or Org is changed to OG throughout. Legends are updated accordingly.

- Supp. Fig. 2 legend: "bivalence increase<s>"

Corrected

- Supp. Fig. 2 legend: should the "Figure 1c" reference be in parentheses at the end?

Corrected

- Supp. Fig. 3 and legend: why is "TAT" now referred to as "TAT11"? Or are they different? And what happened to P4 in the TAT11-(C5)2 construct in panels 3c and 3f? Is this different? In the legend P4 is still mentioned. What are T1 and T2 in 3c and 3f? Presumably tracers, but not otherwise described. Is T2 aka "fluorescently labeled DAT C11" the same as OrgC11 (p. 6) aka OrG-DAT C11 (Supp Fig. 2 legend)?

Tracers are spelled out and labelling in made consistent.

- Supp. Fig. 3 legend: 20 nm concentration.

corrected to nM

- Supp. Fig. 5 shows the titrations of C5 and P4-(C5)2 displacing pre-bound PICK1 from SCSMs. The titration for C5 appears to show a clear decrease from 1000 nM to 10,000 nM, but the summary curve (Fig. 1k) shows almost no inhibition at 10,000 nM. This could reflect the vagaries of a single representative experiment, but it would be good to show a representative experiment that accounts for/is consistent with the trend shown in Fig. 1k.

This has been updated

- Supp. Fig. 8: what is TPD5? TAT-P4-(C5)2 (D for DAT?)? Why another nomenclature? Or is this different?

corrected

- p. 21, line 1: "strenght"

corrected

- Fig. 1 legend: panel (c) is not described; description for panel (c) matches panel (d) etc. In the text, the references are to the correct panels.

Corrected

- Fig. 1d and 1e; why do the legends describe TAT-C5 but TAT11-P4-(C5)₂?

Corrected

- Fig. 1g: "cells expressing... is grown"

Corrected

- Fig. 1k: "pre-complexed"

Corrected

- Fig. 2 legend: the description of panel c "abs280 nm" and "abs544 nm" is confusing. The latter is not the PICK1 profile as described in the sentence, but rather the peptide. Also, the nm should be in the subscript. Finally, the axis label says 546 nm. Which is it? And why is Tat now lower-case in the panels, but not the legend?

Tat is changed to lower case throughout figure and the legend is changed to the correct absorption (546nm). Moreover, the legend for panel c is revised to now read: 'TMR-TAT-P₄-(C₅)₂ (blue (abs_{280nm}), Abs_{max} = 10.7 ml). The elution profile for TMR-TAT-P₄-(C₅)₂ (red (abs_{546nm})) follows the elution profile of PICK1 (abs_{280nm}).'

-Fig 3a: why isn't the panel that contains TMR-C5 (bottom middle) labelled? In the legend, it says "all 20microM" for concentration, but in parentheses after TMR-TAT-C5 it says 5 microM. What concentration(s) is/are used? Fig. 3b says that a 5 nM concentration is used. Is it truly 1000-fold diluted compared to 3a (e.g., to facilitate co-localization)?

The interpretation of the data in Fig 3a bottom panel (middle) is that TMR-C5 is not taken up into the hippocampal neurons nor does it adhere to the surface – even after incubation with 20µM. This is as expected since this peptide has no CPP. We agree that the labeling in the legend was confusing as a result of rearrangements. It now reads: '(a) Representative confocal images of hippocampal neurons showing membrane penetration of TMR-TAT-P₄-(C₅)₂ and TMR-TAT-C₅ (both 5µM) (magenta), but not the control TMR-C₅ (20µM).'

Indeed, figure 3b is done with 5nM peptide and indeed it is to obtain a punctuate/localized distribution enabling colocalization with PICK1 and GluA2. This low concentration is now motivated in the legend which reads: 'Representative confocal images of hippocampal neurons transduced with the viral vector encoding GFP-PICK1 and incubated with a low concentration of TMR-TAT-P₄-(C₅)₂ (5nM) to enable distinct localization in synaptic structures.'

- Fig. 3g and 3h are not referred to in the results.

The figure is now referred to after the original sentence: 'Somewhat surprisingly, TAT-P₄-(C₅)₂ significantly increased constitutive internalization of surface labeled GluA2. Activation of protein kinase C (PKC) by PMA treatment further increased GluA2 internalization, which occluded the effect of TAT-P₄-(C₅)₂ (Figure 4I and J).'

- Fig. 3 title: "TAT-P4-(C5)₂ is ... and compromise<s>..."

Corrected

- Fig. 4 legend: two references to (f), no label for 4g, which is referred to in the text.

The reference to 4g (now 5G) have been removed

Thank you for the submission of your revised manuscript to EMBO Molecular Medicine. We have now received the enclosed reports from the referees that were asked to re-assess it. As you will see the reviewers are now globally supportive and I am pleased to inform you that we will be able to

accept your manuscript pending the following final amendments:

1) Please address the minor text changes commented by referee 1.

***** Reviewer's comments *****

Referee #1 (Remarks for Author):

The revision dramatically improved manuscript. Now, it is clear what authors were aiming to achieve.

Main points:

1. This statement by authors is key point: "The interesting rationale is that this might produce fewer side-effects and a better therapeutic window, simply because one will never fully block glutamatergic transmission but only the subset of receptors under trafficking e.g. during plasticity". This point was not clear in the original submission. Abstract should start from this key point. I could advise to write beginning of abstract following way: "Total blocking of the ionotropic glutamate receptors is a root problem leading to the side effects". Then, "we explored possibility to inhibit these receptors partially by blocking the subset of receptors under trafficking e.g. during plasticity". Then, your text "A cell permeable, highaffinity (~2 nM) peptide inhibitor, Tat-P4-(C5)₂, of the PDZ domain protein PICK1 was developed to achieve this objective". Rest is the same as in the manuscript's abstract.
2. Remaining questions are well addressed. Slice recording from spinal cord of adult mice is indeed challenging. Alternative approach is OK.

Overall, well done. I could suggest changes at the beginning of abstract. It could make objective and novelty of study so much clearer.

Referee #2 (Remarks for Author):

During the revision, the authors have added an impressive list of new experiments to address the comments and concerns raised by reviewers. All concerns raised by this reviewer have been addressed mainly by experiments and by re-writing. The revised manuscript is with significantly improved quality. I support the publication of the manuscript in EMBO Mol Med.

Referee #3 (Remarks for Author):

[this referee has no more comments and is supportive of publication]

3rd revision – authors' response

1st Apr 2020

The authors performed the requested editorial changes.

Corresponding Author Name: Kenneth Lindgaard Madsen

Manuscript Number: EMM-2019-11248-V2